# Weaving in the Clouds: Achieving Synergistic Collaboration among LLM Agents via Federated Learning

## Abstract

Multi-Agent Systems (MAS) powered by Large Language Models (LLMs) have shown immense potential in solving complex, sequential tasks by simulating expert collaboration. However, their reliance on centralized data clashes with real-world privacy constraints and data silos. Conversely, existing privacy-preserving paradigms like Federated Learning (FL) typically ignore the inherent sequential dependencies present in collaborative workflows, leading to suboptimal performance. To bridge this critical gap, we introduce **FedWave**, a novel framework for federated multi-agent collaboration. FedWave empowers LLM-based agents to collaboratively solve complex sequential tasks under strict privacy constraints by employing three core mechanisms: (1) a collaborative Value Chain Layer to model sequential dependencies, enabling efficient local fine-tuning through Federated Learning with LoRA adapters; (2) an intelligent Mixture of Experts (MoE) router at the server level for dynamic, task-aware aggregation of expert knowledge, moving beyond simple averaging; and (3) a final Direct Preference Optimization (DPO) stage to align the model's collaborative outputs with human preferences. Extensive experiments demonstrate that FedWave significantly outperforms both traditional federated learning and centralized multi-agent baselines, effectively achieving synergistic collaboration without compromising data privacy. The codes are available at https://anonymous.4open.science/r/FedWave-111A.

## 1 Introduction

Large Language Models (LLMs) have demonstrated remarkable capabilities across a diverse range of tasks, enabling numerous innovative applications (Webb et al., 2023; Ouyang et al., 2022; Achiam et al., 2023; Team, 2024). Among these, frameworks based on Multi-Agent Systems (MAS) have become particularly prominent for solving complex real-world problems by simulating collaboration among experts from different domains (Zhao et al., 2024; Qian et al., 2025; Li et al., 2024). In scenarios such as business planning (Zhao et al., 2025), financial analysis (Yang et al., 2023), and medical diagnostics (Tang et al., 2024a), MAS effectively model and handle sequential dependencies across different roles, showing immense potential. However, the success of these systems heavily relies on massive, centralized datasets (Wu et al., 2024). This data demand faces two major challenges: first, high-quality public datasets are projected to be exhausted by 2026 (Villalobos et al., 2022); second, vast high-quality data is distributed among different parties, forming "data silos" due to data sovereignty and privacy concerns (Ye et al., 2024; Fan et al., 2023). Therefore, although MAS excel at handling process dependencies, their centralized data assumption overlooks the critical need for privacy preservation in the real world.

Federated Learning (FL) (McMahan et al., 2023) offers a viable solution to this issue. It establishes a privacy-preserving, distributed collaborative framework that allows multiple parties to train a model jointly without sharing their local data, thereby greatly facilitating the secure integration of data value across institutions. Nevertheless, mainstream FL frameworks (Kuang et al., 2024; Yao et al., 2024) often overlook the potential sequential dependencies or causal chains that may exist between different data sources in their design. For instance, in a typical automotive industry chain, design data precedes production data, supply chain data is tightly coupled with the production phase, and quality inspection data serves as the downstream validation for the entire process. Existing federated

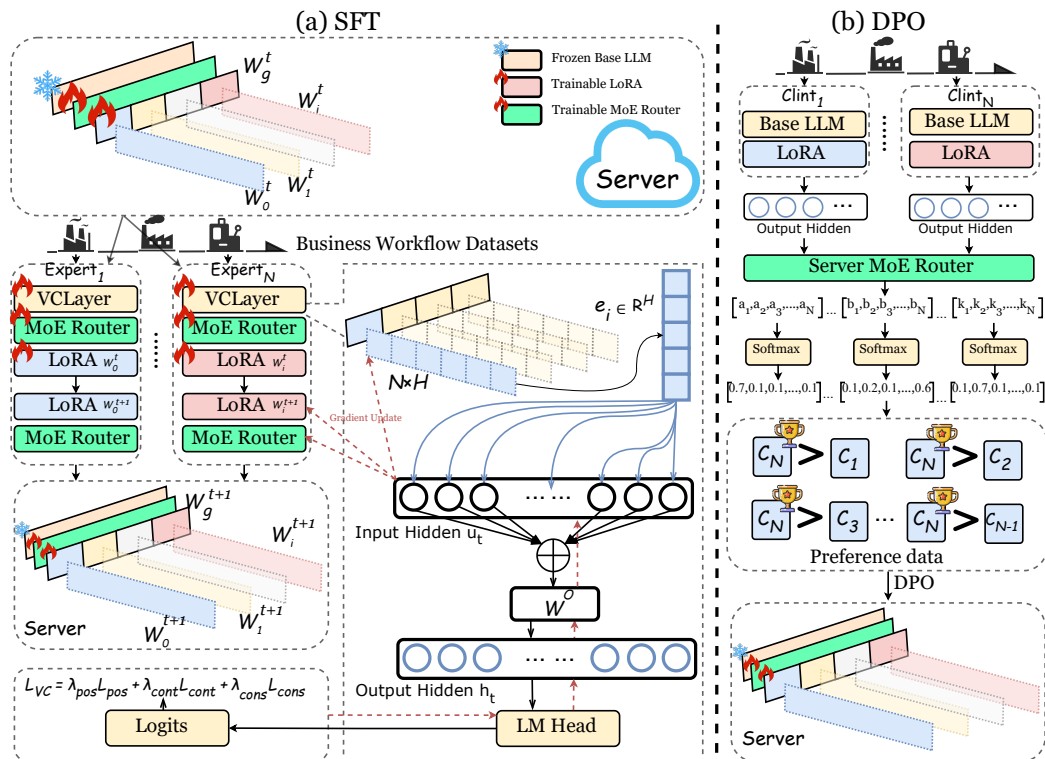

Figure 1: Overview of the FedWave framework. The framework consists of two main phases: (a) The Federated SFT phase, where individual expert agents collaboratively train trainable LoRA adapters, the Value Chain Layer (VCLayer), and a MoE Router on private data. The VCLayer models the workflow structure through a specialized loss function, while the MoE Router learns to dynamically coordinate experts and updates its gradients via backpropagation. Locally updated model parameters are sent to the server for aggregation. (b) The DPO alignment phase, which utilizes the trained MoE Router to automatically generate a preference dataset composed of responses from the most relevant (winning) and less relevant (losing) experts. The final global model is fine-tuned on this dataset using DPO to align its output with high-quality collaborative preferences.

aggregation strategies, typically based on the Independent and Identically Distributed (IID) data assumption (Li et al., 2020; Karimireddy et al., 2021; Hsu et al., 2019), fail to effectively model and leverage this process-aware knowledge embedded in the data, thus limiting the performance of the federated model in complex collaborative tasks with inherent logical sequences.

To address these challenges, we propose FedWave, a novel framework for federated multi-agent collaboration. FedWave empowers LLM-based agents for efficient collaboration while preserving data privacy, integrating distributed knowledge to solve complex sequential tasks. Its core mechanisms are as follows: First, at the client level, we combine FL with the Parameter-Efficient Fine-Tuning (PEFT) technique, LoRA. Each expert agent performs local LoRA-only fine-tuning on the base LLM, minimizing model updates sent to the server. This design significantly enhances communication efficiency and mitigates privacy leakage risks by reducing information exchange. Second, on the server, we designed an innovative Mixture of Experts (MoE) aggregation router. This router discards traditional FL aggregation, acting as an intelligent coordinator that learns to dynamically and selectively weave together knowledge from different expert agents based on the task. Finally, we introduce a Direct Preference Optimization (DPO) (Rafailov et al., 2023) stage to align the aggregated global model using preference data on collaborative outcomes. This ensures the model generates high-quality, collaborative outputs that align with human expectations. Extensive experiments demonstrate that our framework outperforms existing federated learning and multi-agent baselines.

The main contributions of this paper can be summarized as follows:

- **Privacy-Preserving Collaboration for Workflow Tasks:** We introduce FedWave, the first framework enabling LLM agents to solve complex, sequential workflows across data silos. Its novel Value Chain Layer, within a Federated Learning paradigm, models inter-agent dependencies to facilitate structured, privacy-preserving collaboration.
- **Dynamic Knowledge Aggregation:** We designed a aggregation mechanism centered on MoE router. This shifts from static federated averaging to dynamic, task-aware knowledge fusion, allowing the global model to selectively leverage the most relevant expertise.
- **Extensive Experimental Validation:** Comprehensive experiments on business workflow datasets validate FedWave's superiority. It significantly outperforms federated learning baselines and achieves competitive results against centralized multi-agent systems.

## 2 RELATED WORK

### 2.1 FEDERATED LEARNING

Federated Learning (FL) is a privacy-preserving paradigm for collaborative training on decentralized data (Kairouz et al., 2021). While the foundational FedAvg algorithm (McMahan et al., 2023) is effective, its performance often degrades with Non-IID data, prompting extensive research into solutions. These improvements typically mitigate data heterogeneity through client-side optimizations like FedProx (Li et al., 2020) and SCAFFOLD (Huang et al., 2024), or server-side aggregation refinements such as FedNova (Wang et al., 2020) and FedAvgM (Hsu et al., 2019). However, existing works primarily address statistical heterogeneity, overlooking the critical issue of Sequential Dependency in process-based tasks. Their static aggregation methods treat clients as independent contributors, failing to dynamically fuse knowledge based on the task's inherent structure. In contrast, our FedWave framework fundamentally changes this paradigm by enabling the model to understand and leverage these sequential relationships among clients.

### 2.2 MULTI-AGENT COLLABORATION

The rise of Large Language Models (LLMs) has significantly advanced the development of Multi-Agent Systems, establishing them as a powerful paradigm for solving complex problems (Akata et al., 2023; Guo et al., 2024; Hao et al., 2023). These systems accomplish tasks by simulating collaboration among multiple agents, each with distinct roles or capabilities. Existing works have explored diverse collaboration models: some frameworks leverage discussion and debate (Du et al., 2024; Chen et al., 2024; Xiong et al., 2023) to refine and enhance reasoning abilities, while others construct hierarchical or sequential pipeline structures (Zhang et al., 2024; Zhao et al., 2024; 2025). In these structures, agents process information progressively according to a predefined workflow (such as a business workflow), effectively addressing complex sequential tasks in domains like long-text processing and medical diagnostics (Tang et al., 2024b; Sun et al., 2023). However, these frameworks' reliance on centralized data access fundamentally conflicts with real-world, privacy-sensitive applications where data is siloed. This reveals a critical research gap: enabling complex, sequentially dependent agent collaboration in a decentralized, privacy-preserving setting. Our work, FedWave, directly addresses this challenge by merging the collaborative power of multi-agent systems with the privacy guarantees of federated learning.

### 2.3 MIXTURE OF EXPERTS AND DYNAMIC AGGREGATION

The Mixture of Experts (MoE) architecture enhances model scalability through conditional computation, where a router dynamically selects specialized 'expert' subnetworks for each input (Jacobs et al., 1991; Fedus et al., 2022). After its revival in modern deep learning (Shazeer et al., 2017), this paradigm has been instrumental in scaling Large Language Models (LLMs) to trillion-parameter scales while maintaining computational efficiency (Lepikhin et al., 2020; Jiang et al., 2024). This principle inspires Dynamic Aggregation in Federated Learning (FL), moving beyond the static, one-size-fits-all approach of methods like FedAvg (McMahan et al., 2017). While Personalized FL (PFL) adapts models for clients (Fallah et al., 2020; Arivazhagan et al., 2019; T Dinh et al., 2020; Li et al.,

2021), it primarily addresses statistical heterogeneity rather than collaborative, sequential relationships. However, a critical gap remains, as existing work has not integrated MoE's dynamic routing into FL aggregation to specifically solve for Sequential Dependency among clients. Our FedWave framework is the first to introduce an MoE router as the core of federated aggregation, transforming it into an intelligent coordinator that can orchestrate expert knowledge for complex workflows.

# 3 METHODS

## 3.1 FEDERATED FINE-TUNING WITH THE VALUE CHAIN LAYER

The first phase of our framework fine-tunes a base LLM in a federated setting where each client, an expert agent (e.g., design, manufacturing), holds a private dataset for their role. To model the sequential relationships between these experts, we introduce the Value Chain Layer (VCLayer). As shown in Figure 1 (a), each expert agent $i$ has a frozen base LLM, a trainable LoRA adapter ($W_i$), and our VCLayer. This lightweight, pluggable module processes the hidden states from the LoRA-adapted model, making them aware of the agent's role and position in the workflow.

Given the final hidden states $H \in \mathbb{R}^{L \times d}$ from the base LLM (where $L$ is sequence length, $d$ is hidden dimension), the VCLayer applies a stage-specific transformation. For an agent at stage $i$, this is a specialized multi-head attention mechanism:

$$H'_i = \text{Attention}(HW_{q,i}, HW_{k,i}, HW_{v,i})W_{o,i} \tag{1}$$

where $W_{q,i}, W_{k,i}, W_{v,i} \in \mathbb{R}^{d \times d_{vc}}$ and $W_{o,i} \in \mathbb{R}^{d_{vc} \times d}$ are trainable, stage-specific projection matrices. This allows each expert to focus on task-relevant aspects of the input. The resulting state $H'_i$ is then passed to the LM head to produce logits. A key innovation is the collaborative loss, $\mathcal{L}_{VC}$, which guides the VCLayer to learn the workflow structure. During local training for agent $i$, the total loss combines the standard Supervised Fine-Tuning (SFT) loss with our collaborative loss:

$$\mathcal{L}_{\text{total}}^{(i)} = \mathcal{L}_{\text{SFT}}^{(i)} + \gamma \mathcal{L}_{VC}^{(i)} \tag{2}$$

Here, $\gamma$ is a balancing hyperparameter. $\mathcal{L}_{\text{SFT}}$ is the conventional cross-entropy loss for next-token prediction. $\mathcal{L}_{VC}$ comprises three terms that enforce the value chain's relational structure:

$$\mathcal{L}_{VC} = \lambda_{\text{pos}}\mathcal{L}_{\text{pos}} + \lambda_{\text{cont}}\mathcal{L}_{\text{cont}} + \lambda_{\text{cons}}\mathcal{L}_{\text{cons}} \tag{3}$$

where $\lambda_{\text{pos}}$, $\lambda_{\text{cont}}$, and $\lambda_{\text{cons}}$ are weighting coefficients.

**Positional Loss ($\mathcal{L}_{\text{pos}}$):** This loss enforces a geometric arrangement of experts in an embedding space, reflecting their sequential order. It is based on the similarity between learnable embeddings $\{e_0, ..., e_{N-1}\}$ for each stage:

$$\mathcal{L}_{\text{pos}} = \sum_{i=0}^{N-2} \left( \cos(e_i, e_{i+1}) - T_{\text{pos}}^{(i)} \right)^2 \tag{4}$$

where $T_{\text{pos}}^{(i)}$ is a target cosine similarity, encouraging adjacent experts to be closer.

**Continuity Loss ($\mathcal{L}_{\text{cont}}$):** This loss promotes a smooth transition of knowledge by ensuring adjacent experts learn similar functions. It operates on the VCLayer's projection matrices:

$$\mathcal{L}_{\text{cont}} = \sum_{i=0}^{N-2} \left( \cos(\text{vec}(W_{q,i}), \text{vec}(W_{q,i+1})) - T_{\text{cont}} \right)^2 \tag{5}$$

where $\text{vec}(\cdot)$ is the vectorization operator and $T_{\text{cont}}$ is a high target similarity value, encouraging the attention mechanisms of consecutive stages to be functionally alike.

**Consistency Loss ($\mathcal{L}_{\text{cons}}$):** This loss ensures a coherent solution progression by aligning an expert's output representation with its predecessor's for the same input. For an agent at stage $i > 0$:

$$\mathcal{L}_{\text{cons}}^{(i)} = \left( \cos(f_i(\bar{H}), f_{i-1}(\bar{H})) - T_{\text{cons}} \right)^2 \tag{6}$$

where $f_i(\bar{H})$ is the mean-pooled output hidden state from the VCLayer of agent $i$ for input $H$.

## 3.2 DYNAMIC AGGREGATION WITH MoE ROUTING

To overcome static aggregation limitations (e.g., FedAvg), we introduce a trainable Mixture of Experts (MoE) Router. This task-aware coordinator dynamically determines each expert's contribution based on the input prompt. As a shared global component, it is co-trained with the experts' LoRA adapters and VCLayers. The MoE Router, $G(\cdot)$, is a lightweight MLP. For any input, it computes a representation by mean-pooling the base LLM's hidden states, $\bar{H} = \frac{1}{L} \sum_{l=1}^{L} H_l$. It then processes $\bar{H}$ to produce logits $z \in \mathbb{R}^N$ over the $N$ experts, which are converted to a probability distribution via softmax:

$$\alpha = \text{softmax}(G(\bar{H})) \tag{7}$$

where $\alpha_i$ is the routing weight for expert $i$. The MoE Router is trained implicitly via a loss weighting mechanism during local SFT. We modulate each expert's SFT loss for a sample by its assigned routing weight. For expert $i$, the loss $\mathcal{L}_{\text{SFT}}^{(i)}$ is multiplied by $\alpha_i$, strengthening the gradient signal when the router correctly assigns a high weight to the appropriate expert.

We add two auxiliary losses to regularize the router. A **load balancing loss**, $\mathcal{L}_{\text{balance}}$, encourages even expert utilization over a batch to prevent specialization collapse, and is formulated as the variance of expert utilization. An **entropy-based confidence loss**, $\mathcal{L}_{\text{entropy}}$, penalizes uncertain routing decisions to encourage sparse, confident weights. The total local loss for agent $i$ is thus:

$$\mathcal{L}_{\text{total}}^{(i)} = \alpha_i \mathcal{L}_{\text{SFT}}^{(i)} + \gamma \mathcal{L}_{VC}^{(i)} + \delta_1 \mathcal{L}_{\text{balance}} + \delta_2 \mathcal{L}_{\text{entropy}} \tag{8}$$

where $\delta_1$ and $\delta_2$ are hyperparameters. This composite loss enables end-to-end training of the LoRA adapters, VCLayers, and the MoE Router. After each local training round, the router's updated weights are aggregated on the server, similar to FedAvg:

$$W_{\text{router}}^{t+1} = \sum_{i \in S_t} \frac{n_i}{n} W_{\text{router},i}^{t+1} \tag{9}$$

## 3.3 PREFERENCE ALIGNMENT WITH DIRECT PREFERENCE OPTIMIZATION

After the federated SFT phase, the aggregated model has expert knowledge and a foundational workflow understanding. The final phase (Figure 1 (b)) refines the model's output by aligning it with human preferences for quality and coherence using Direct Preference Optimization (DPO) (Rafailov et al., 2023). This phase begins by using the final SFT model, $\pi_{\text{SFT}}$, to automatically generate a preference dataset, $\mathcal{D}_{\text{pref}} = \{(x, y_w, y_l)\}$. The co-trained MoE Router's intelligence is leveraged to create these preference pairs. For each prompt $x$, the following steps are taken:

1. The MoE router within $\pi_{\text{SFT}}$ computes routing weights $\alpha$ and ranks the $N$ experts based on their relevance to the prompt.

2. The top-ranked expert, $c_w = \arg\max_i \alpha_i$, is selected to generate the winning (chosen) response, $y_w$.

3. A lower-ranked expert, $c_l$, is selected to generate the losing (rejected) response, $y_l$.

4. The tuple $(x, y_w, y_l)$ is added to the preference dataset $\mathcal{D}_{\text{pref}}$.

This automated process uses the router's expertise to create a large-scale dataset favoring outputs from the most contextually appropriate expert. The aggregated model $\pi_{\text{SFT}}$ is then fine-tuned on $\mathcal{D}_{\text{pref}}$ using DPO. DPO directly optimizes the model for preferences without a separate reward model. The policy model, $\pi_\theta$, is initialized from $\pi_{\text{SFT}}$, and a frozen copy of $\pi_{\text{SFT}}$ serves as the reference model, $\pi_{\text{ref}}$. DPO's objective is to maximize the likelihood of preferred responses $y_w$ and minimize that of rejected responses $y_l$, constrained by a penalty term preventing large deviations from the reference model. The loss function is:

$$\mathcal{L}_{\text{DPO}}(\pi_\theta; \pi_{\text{ref}}) = -\log \sigma \left( \beta \log \frac{\pi_\theta(y_w|x)}{\pi_\theta(y_l|x)} - \beta \log \frac{\pi_{\text{ref}}(y_w|x)}{\pi_{\text{ref}}(y_l|x)} \right) \tag{10}$$

where $\sigma$ is the logistic function and $\beta$ controls the preference strength. Minimizing this loss over the preference dataset aligns the policy model $\pi_\theta$ with the collaborative logic from the SFT phase. This final step yields the fully optimized FedWave model, adept at specialized tasks and high-quality, human-preferred collaboration.

Table 1: Performance comparison of FedWave and baselines across three BizWorkflow datasets and three different backbone models. The best scores for each metric are highlighted in **bold**.

| Baselines | Automotive | | | E-commerce | | | Pharmaceutical | | |
|---|---|---|---|---|---|---|---|---|---|
| | **BS-F** | **Meteor** | **Rouge-L** | **BS-F** | **Meteor** | **Rouge-L** | **BS-F** | **Meteor** | **Rouge-L** |
| *Qwen2-7B* | | | | | | | | | |
| FedAvg | 71.11 | 23.35 | 22.18 | 79.17 | 43.29 | 38.68 | 77.12 | 26.76 | 28.54 |
| FedAvgM | 70.66 | 22.02 | 21.74 | 79.51 | 43.09 | 39.23 | 76.87 | 25.57 | 27.51 |
| FedProx | 71.36 | 23.58 | 22.53 | 79.23 | 43.53 | 38.84 | 77.18 | 27.08 | 28.65 |
| FedAdam | 71.39 | 23.74 | 21.74 | 78.78 | 41.01 | 37.07 | 76.41 | 26.07 | 27.29 |
| FedYogi | 71.49 | 24.18 | 22.48 | 78.54 | 40.31 | 36.81 | 76.00 | 24.59 | 26.11 |
| Scaffold | 71.26 | 23.53 | 22.49 | 79.04 | 42.32 | 38.50 | 77.06 | 27.09 | 28.68 |
| **FedWave** | **71.90** | **40.35** | **23.46** | **80.17** | **48.28** | **39.63** | **78.34** | **29.82** | **31.89** |
| *Llama2-7B* | | | | | | | | | |
| FedAvg | 69.43 | 17.41 | 20.98 | 72.96 | 16.45 | 22.66 | 70.61 | 10.10 | 14.98 |
| FedAvgM | 65.62 | 14.32 | 16.65 | 73.11 | 18.13 | 23.73 | 70.95 | 11.24 | 15.15 |
| FedProx | 69.26 | 17.08 | 20.74 | 72.82 | 16.30 | 22.52 | 70.92 | 10.27 | 15.06 |
| FedAdam | 68.90 | 17.03 | 19.55 | 70.83 | 13.49 | 19.88 | 68.79 | 8.75 | 13.34 |
| FedYogi | 69.12 | 17.20 | 19.99 | 70.42 | 13.15 | 19.40 | 68.36 | 8.31 | 12.94 |
| Scaffold | 69.25 | 16.78 | 20.64 | 72.95 | 16.62 | 22.72 | 70.97 | 10.39 | 15.25 |
| **FedWave** | **70.08** | **18.55** | **22.12** | **74.95** | **20.80** | **26.30** | **74.78** | **13.38** | **19.73** |
| *Llama3-8B* | | | | | | | | | |
| FedAvg | 58.62 | 10.73 | 9.55 | 68.81 | 26.25 | 25.29 | 61.34 | 14.17 | 14.49 |
| FedAvgM | 56.03 | 7.33 | 6.76 | 64.61 | 24.05 | 21.19 | 61.84 | 14.24 | 14.38 |
| FedProx | 58.44 | 10.50 | 9.27 | 68.52 | 27.16 | 25.66 | 61.79 | 14.50 | 15.06 |
| FedAdam | 51.47 | 2.67 | 2.25 | 57.22 | 18.28 | 14.65 | 58.08 | 12.16 | 11.49 |
| FedYogi | 52.78 | 4.17 | 3.66 | 57.37 | 18.16 | 14.77 | 58.20 | 11.68 | 11.05 |
| Scaffold | 59.15 | 11.31 | 10.21 | 68.12 | 25.65 | 24.60 | 61.62 | 14.85 | 14.97 |
| **FedWave** | **70.27** | **24.83** | **17.47** | **79.66** | **44.72** | **38.33** | **77.95** | **24.02** | **28.40** |

# 4 EXPERIMENTS

Our experiments evaluate FedWave to address three questions: **(1)** How does its collaborative performance on sequential tasks compare to standard federated learning? **(2)** How does the privacy-preserving FedWave perform against a centralized multi-agent system with full data access? **(3)** Which design elements are most critical to its success?

## 4.1 SETTINGS

**Datasets and evaluation metrics.** We evaluate our framework on the MSCoRe benchmark (Lei et al., 2025), which is specifically designed for multi-stage collaborative reasoning. It provides three challenging datasets with complex, sequential tasks representing distinct business workflows: **E-commerce**, **Pharmaceutical**, and **Automotive**. To comprehensively assess the quality of the generated outputs, we employ a diverse suite of metrics beyond simple lexical overlap. This includes ROUGE-1/2/L (Lin, 2004) for lexical content, BLEU-4 (Papineni et al., 2002) and GLEU (Wu et al., 2016) for fluency, and crucially, METEOR (Banerjee & Lavie, 2005) and BERTScore (Zhang et al., 2019) to evaluate deeper semantic fidelity and contextual relevance.

**Baselines.** To validate our framework's effectiveness, we compare it against two baseline categories. For federated learning, we adapt widely-recognized algorithms: the foundational FedAvg (McMahan et al., 2017); FedProx (Li et al., 2020) with its proximal term to mitigate heterogeneity; FedAvgM (Hsu et al., 2019), which adds server-side momentum; SCAFFOLD (Karimireddy et al., 2021) for client-drift correction; and the adaptive optimizers FedAdam and FedYogi (Reddi et al., 2020). For multi-agent systems, we benchmark against centralized meth-

Table 2: Performance comparison of FedWave against centralized multi-agent baselines on the Qwen2-7B backbone. FedWave operates in a decentralized, privacy-preserving setting, while the baselines have access to the full, centralized dataset. The best scores for each metric are highlighted in **bold**.

| Dataset | Baselines | BS-F | GLEU | BLEU-4 | ROUGE-1 | ROUGE-2 | ROUGE-L |
|---------|-----------|------|------|--------|---------|---------|---------|
| Automotive | PMC (Zhang et al., 2025) | 65.81 | 13.25 | 5.24 | 26.44 | 4.51 | 18.37 |
| | MedAgents (Tang et al., 2024b) | 64.94 | 12.47 | 4.81 | 23.68 | 4.07 | 18.34 |
| | Debate(long) (Du et al., 2024) | 65.56 | 12.92 | 6.49 | 25.49 | 5.91 | 18.03 |
| | Debate(short) (Du et al., 2024) | 65.32 | 12.66 | 6.30 | 25.39 | 6.02 | 17.81 |
| | CoA (Zhang et al., 2024) | 70.77 | **22.30** | 14.41 | 33.99 | 11.15 | **24.01** |
| | **FedWave (Ours)** | **71.47** | 20.42 | **15.44** | **35.11** | **12.19** | 22.47 |
| E-commerce | PMC (Zhang et al., 2025) | 70.88 | 24.01 | 19.73 | 33.47 | 11.45 | 29.91 |
| | MedAgents (Tang et al., 2024b) | 69.57 | 17.48 | 12.60 | 33.62 | 10.77 | 28.25 |
| | Debate(long) (Du et al., 2024) | 72.89 | 20.81 | 15.46 | 36.43 | 13.12 | 28.04 |
| | Debate(short) (Du et al., 2024) | 72.93 | 20.85 | 15.56 | 37.11 | 13.51 | 28.23 |
| | CoA (Zhang et al., 2024) | 79.20 | 38.08 | 34.23 | 50.02 | 25.35 | 39.02 |
| | **FedWave (Ours)** | **80.17** | **42.60** | **39.56** | **51.04** | **26.22** | **39.63** |
| Pharmaceutical | PMC (Zhang et al., 2025) | 68.55 | 17.34 | 11.67 | 28.16 | 7.52 | 25.93 |
| | MedAgents (Tang et al., 2024b) | 65.96 | 10.60 | 4.46 | 25.91 | 6.17 | 21.13 |
| | Debate(long) (Du et al., 2024) | 61.14 | 9.58 | 5.33 | 16.18 | 4.69 | 12.23 |
| | Debate(short) (Du et al., 2024) | 70.57 | 15.78 | 9.11 | 31.76 | 9.60 | 23.03 |
| | CoA (Zhang et al., 2024) | 77.26 | **29.92** | 18.18 | 42.38 | 19.93 | **33.70** |
| | **FedWave (Ours)** | **78.34** | 26.27 | **19.22** | **43.24** | **20.00** | 31.89 |

ods that operate on an aggregated dataset: the hierarchical PMC (Zhang et al., 2025), discussion-based MedAgents (Tang et al., 2024b), chain-based CoA (Zhang et al., 2024), and Debate (Du et al., 2024), which enables collaboration through argumentative discourse by modulating agent confidence through varying debate durations.

**Details.** Unless otherwise specified, all our experiments use 7B/8B parameter scale LLMs (Qwen2-7B, Llama2-7B, Llama3-8B) as the base models, which are quantized to 8-bit for computational efficiency. Our framework consists of 4 expert clients, with all clients participating in each communication round. For the federated fine-tuning phase, we run for 20 communication rounds. In each round, every client trains locally for 10 steps using the AdamW optimizer with a batch size of 4. We apply a cosine learning rate schedule, decaying from an initial value of $5 \times 10^{-5}$ to $1 \times 10^{-6}$ over the rounds. The maximum sequence length is set to 2048. For the PEFT technique, we use LoRA with a rank of 32 and a scalar alpha of 64. For our proposed FedWave components, the MoE router selects the top-2 experts ($k = 2$), and the key hyperparameters for the VCLayer losses are set as $\lambda_{pos} = 0.1$, $\lambda_{cont} = 0.1$, $\lambda_{cons} = 0.2$, and $\gamma = 0.5$. For the final preference alignment phase, the aggregated model is fine-tuned using DPO for 10 epochs with a learning rate of $1 \times 10^{-5}$ and a $\beta$ of 0.1. All experiments are conducted on NVIDIA A40 GPUs. We use the Alpaca (Taori et al., 2023) template to format the instructions.

## 4.2 Collaborative Performance of FedWave Compared to Federated Learning

We present the main experimental results in Table 1, comparing our FedWave framework against six federated learning baselines across three business workflow datasets and three LLM backbones. The results clearly demonstrate the consistent superiority of our FedWave framework, as it significantly outperforms all baseline methods across all evaluation metrics, datasets, and backbone models. The performance improvement is particularly pronounced in metrics like Meteor and BERTScore-F, which measure semantic coherence and fluency. For instance, on the Automotive dataset with the Qwen2-7B backbone, FedWave achieves a Meteor score of 40.35, substantially improving upon the best baseline score of 24.18 (FedYogi). This indicates that by explicitly modeling sequential dependencies and dynamically aggregating expert knowledge, FedWave generates outputs that are not only more accurate but also more semantically coherent. Furthermore, this superiority is not confined to a specific model architecture.

Table 3: Ablation study of the key components of FedWave on the Automotive dataset using the Llama3-8B backbone.

| Method | BS-F | Meteor | Rouge-L |
|---|---|---|---|
| **FedWave (Full Model)** | **70.27** | **24.83** | **17.47** |
| *- Ablation Variants -* | | | |
| w/o DPO | 66.55 | 18.40 | 17.14 |
| w/o MoE Router | 69.00 | 23.65 | 16.99 |
| w/o VCLayer | 69.07 | 23.96 | 16.21 |
| *- Baseline -* | | | |
| SFT Only (FedAvg) | 58.62 | 10.73 | 9.55 |

Table 4: Performance of the FedWave framework when integrated with different federated optimization algorithms. Experiments are conducted with the Llama-8B backbone.

| Aggregation Algorithm | BS-F | Meteor | Rouge-L |
|---|---|---|---|
| FedWave (Default) | 66.55 | 18.40 | 17.14 |
| + FedAvgM | 70.59 | 23.43 | 21.73 |
| + FedProx | **70.86** | 23.71 | **22.20** |
| + FedAdam | 70.68 | 23.68 | 21.89 |
| + FedAdagrad | 70.71 | **23.79** | 21.97 |
| + FedYogi | 70.35 | 23.42 | 21.46 |

## 4.3 COLLABORATIVE PERFORMANCE OF FEDWAVE COMPARED TO MULTI-AGENT

We further benchmark FedWave against several centralized Multi-Agent baselines, as shown in Table 2. Crucially, while baselines operate with full data access, FedWave performs in a decentralized, privacy-preserving setting. Despite this challenging condition, our framework demonstrates highly competitive or even superior performance. On the E-commerce dataset, FedWave surpasses all centralized methods across every reported metric. On the Automotive and Pharmaceutical datasets, it achieves state-of-the-art results on key metrics such as BERTScore-F and ROUGE, proving its ability to generate high-quality, semantically rich outputs.

## 4.4 KEY DESIGN FACTORS AND HYPERPARAMETER INFLUENCE IN FEDWAVE

**Ablation Study.** The ablation study results in Table 3 reveal the powerful synergistic effect of our framework's components. While removing any single component—the DPO stage, MoE Router, or VCLayer—causes a noticeable performance degradation, the decline is not catastrophic as the remaining parts partially compensate to maintain a degree of collaborative intelligence. For instance, without the MoE Router, the VCLayer still ensures the agents learn their sequential roles. However, when all three components are removed (reverting to the 'SFT Only (FedAvg)' baseline), the collaborative intelligence system collapses, causing a drastic drop across all metrics (e.g., Meteor falls from 24.83 to 10.73). This substantial gap demonstrates that our components have a multiplicative, not merely additive, effect. It is the integrated combination of explicit workflow modeling (VCLayer), dynamic knowledge aggregation (MoE Router), and preference alignment (DPO) that collectively enables FedWave's superior performance.

**Hyper-parameter Sensitivity Analysis.** We conduct a comprehensive analysis to evaluate the sensitivity of FedWave to its key hyperparameters, demonstrating the framework's robustness.

**VCLayer Loss Components.** As illustrated in Figure 2, we first vary the individual weights for the Positional Loss ($\lambda_{pos}$), Continuity Loss ($\lambda_{cont}$), and Consistency Loss ($\lambda_{cons}$). For both $\lambda_{pos}$ and

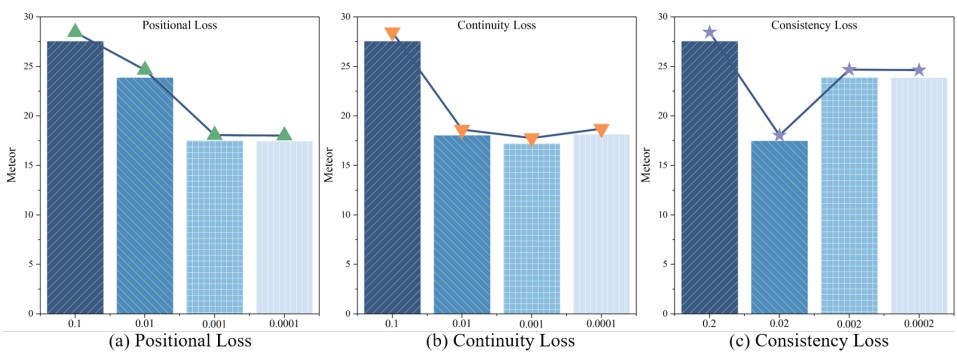

Figure 2: Sensitivity analysis for the weights of the collaborative loss components in the VCLayer.

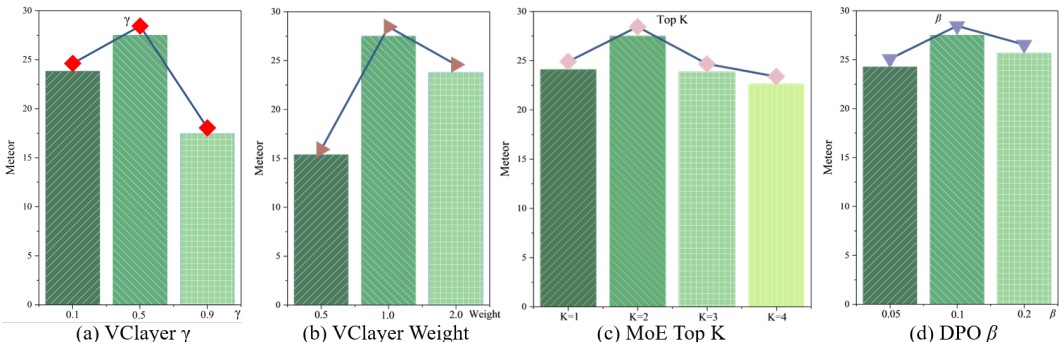

Figure 3: Sensitivity analysis for core hyperparameters of FedWave: (a) Balancing coefficient $\gamma$ for SFT and VC losses; (b) Overall scaling factor for VCLayer weights; (c) MoE router's top_k value; and (d) DPO's beta value.

$\lambda_{cont}$, the performance, measured by Meteor, peaks at a weight of 0.1 and gracefully degrades as the weight decreases, confirming their significant contribution to modeling the workflow structure. The framework also shows robustness to changes in $\lambda_{cons}$, maintaining stable performance across a range of values.

**Other Core Hyperparameters.** We further analyze several critical hyperparameters in Figure 3. We study the balancing coefficient $\gamma$ (Figure 3 (a)), which trades off between the SFT and VCLayer losses, finding that $\gamma = 0.5$ yields an optimal balance. Our analysis of an overall scaling factor for the VCLayer weights (Figure 3 (b)) shows performance peaks at the default scaling of 1.0, confirming our weights are well-calibrated. The study of MoE top_k (Figure 3 (c)) reveals that performance peaks at k=2, suggesting that activating a small, focused group of experts is most effective. Finally, the analysis of DPO $\beta$ (Figure 3 (d)) confirms that a value of 0.1 provides the best alignment without significant deviation from the base model. Overall, these results demonstrate that while our proposed components are crucial, FedWave is not overly sensitive to their precise hyperparameter values, highlighting its stability and reliability.

## 4.5 ANALYSIS ON FEDERATED AGGREGATION ALGORITHMS

To assess the compatibility and modularity of our framework, we integrated FedWave with several advanced federated optimization algorithms, replacing the default FedAvg-based aggregation. The results, presented in Table 4, show that all tested optimizers yield a substantial performance improvement over the default configuration. Notably, algorithms like FedProx and those with adaptive optimization (FedAdagrad, FedAdam) achieve the higher scores. This suggests that the structured, non-IID environment created by our VCLayer and MoE router benefits significantly from optimizers designed to handle client drift and heterogeneity. FedProx, with its regularization term, likely prevents the specialized expert models from diverging too far from the global consensus, while adaptive methods better navigate the complex loss landscape.

## 5 CONCLUSION

In this paper, we introduced FedWave, a novel federated multi-agent collaboration framework designed to address the critical challenge of solving complex sequential tasks across decentralized, privacy-sensitive data silos. By integrating a collaborative VCLayer, a dynamic MoE router for intelligent aggregation, and a final DPO stage for preference alignment, our framework successfully bridges the gap between the collaborative capabilities of multi-agent systems and the privacy guarantees of federated learning. Our extensive experiments demonstrate that FedWave not only significantly outperforms standard federated learning baselines but also achieves performance competitive with, and often superior to, centralized multi-agent systems that have unrestricted data access.

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

## A  QUALITATIVE ANALYSIS: A CASE STUDY

To provide a more intuitive understanding of the performance differences, we present a qualitative case study on a representative task from the Automotive workflow dataset. We prompted the models with a complex, two-part request that requires both marketing creativity (Expert 1's domain) and supply chain knowledge (Expert 2's domain). The results, shown in Figure 4, highlight the distinct collaborative capabilities of each approach.

The analysis of the outputs in Figure 4 reveals the clear superiority of our proposed framework. The **FedAvg**, lacking any mechanism for structured collaboration, produces a convoluted and generic response. It struggles to differentiate between the distinct tasks of marketing and supply chain planning, mixing concepts and failing to provide the specific, actionable details required by either role. This output exemplifies the shortcomings of simple model averaging, which dilutes expert knowledge into a generalized but ultimately ineffective model.

In contrast, the **Centralized MAS** baseline, with full data access, performs well. It correctly separates the two sub-tasks and provides relevant, technically sound details for each, such as focusing on the core chassis components. This demonstrates the effectiveness of the multi-agent paradigm when privacy is not a concern.

Most importantly, **FedWave** achieves a remarkable level of performance that is highly competitive with the centralized approach, despite operating under strict privacy constraints. The output demonstrates a clear and seamless transition between the two roles. The marketing announcement is creative, while the supply chain requirements are precise and technical. Crucially, the generated details are not only plausible but also qualitatively different from the centralized model's output—for instance, specifying innovative components like 'BladeCell' batteries and 'WaveDrive' motors, and focusing on sustainable interior textiles. This suggests that `FedWave` successfully synthesizes specialized knowledge learned from distributed data sources, a feat not possible with simple averaging. This successful role-switching is a direct result of our **MoE router** dynamically identifying the relevant expert for each part of the prompt. Furthermore, the high quality and coherence of the combined output showcase the effectiveness of the **VCLayer** in maintaining a logical flow and the **DPO** stage in refining the final generation to align with high-quality, human-like collaborative solutions. This case study compellingly illustrates that FedWave successfully instills genuine collaborative intelligence in a decentralized environment.

| | |
|---|---|
| | **Input Prompt**

*"Based on the latest market analysis indicating a strong demand for sustainable family vehicles, draft a marketing announcement for our new electric SUV, the 'EcoVoyager'. Then, outline the key component requirements for the initial production run that the supply chain team needs to source."* |
| **Model** | **Generated Output** |
| **Centralized MAS** (Full Data Access) | ***Marketing Announecment:***

*Introducing the all-new EcoVoyager! Experience the future of family adventure with zero emissions. Designed for the conscious modern family with an exceptional range and advanced safety features. Pre-orders open this fall. #EcoVoyager #ElectricSUV #SustainableDriving*

***Supply Chain Requirements:***

*For the initial production run of 10,000 units, the key components to source are:*

   *1. Battery Packs: High-capacity lithium-ion battery packs meeting our long-range specifications.*
   *2. Electric Motors: High-efficiency dual-axle permanent magnet motors for the all-wheel-drive system.*
   *3. Infotainment System: The latest-generation central touchscreen with our proprietary OS.*
   *4. Chassis Components: Lightweight, high-strength aluminum alloy for the vehicle's unibody frame.* |
| **FedAvg** | ***Marketing and Supply Chain Plan:***

*Unveil your family's next chapter with the EcoVoyager! Fusing sustainable luxury with thrilling performance, our all-electric SUV is here. Featuring a panoramic sunroof and an interior crafted from premium eucalyptus fiber, the EcoVoyager delivers an impressive range on a single charge. Drive the change. #EcoVoyager #FutureIsElectric #FamilySUV* |
| **FedWave** (Privacy-Preserving) | ***Marketing Announecment:***

*Unveil your family's next chapter with the EcoVoyager! Fusing sustainable luxury with thrilling performance, our all-electric SUV is here. Featuring a panoramic sunroof and an interior crafted from premium eco-friendly materials, the EcoVoyager delivers a 300+ mile range on a single charge. Drive the change. #EcoVoyager #FutureIsElectric #FamilySUV*

***Key Component Requirements:***

*Supply chain priority sourcing for the initial 10,000-unit production run of the EcoVoyager:*

   *1. Battery System: High-density 'BladeCell' battery modules with advanced integrated thermal management.*
   *2. Drivetrain: Next-generation 'WaveDrive' axial flux synchronous motors for superior power delivery (front and rear).*
   *3. Central Computing Unit: The 'VisionCore AI' automotive-grade processor for infotainment and driver-assist features.*
   *4. Sustainable Textiles: Sourcing of certified eucalyptus fiber textiles for all seating and interior trim..* |

Figure 4: Qualitative comparison of model outputs for a multi-stage task. The prompt requires the model to first act as a marketing expert and then as a supply chain analyst.

