# OpenReview forum: "Weaving in the Clouds: Achieving Synergistic Collaboration among LLM Agents via Federated Learning"
_ICLR.cc/2026/Conference — Submitted to ICLR 2026_

### Official Review · Reviewer_B8pN · 2025-10-26

**Soundness:** 2
**Presentation:** 2
**Contribution:** 3
**Rating:** 4
**Confidence:** 3

**Summary:**

This paper proposes a novel framework, FedWave, which integrates federated learning with large language model (LLM)-based multi-agent collaboration. The framework captures sequential dependencies through a Value Chain Layer (VCLayer), achieves cross-task knowledge sharing via Mixture-of-Experts (MoE) dynamic aggregation, and aligns outputs with human preferences using Direct Preference Optimization (DPO). The paper is well-structured, and the experiments are comprehensive. The work is innovative, technically solid, and represents a cutting-edge exploration in federated learning and LLM collaboration. However, the theoretical analyses are slightly lacking, and the interpretability needs to be strengthened, so the paper has a strong potential for acceptance if it can be further improved.

**Strengths:**

1. Originality and Novelty: This is the first work to introduce sequential dependency modeling into a federated learning framework, formalizing multi-agent collaboration as a distributed optimization problem.

2. Completeness of System Design: The proposed three-stage architecture, structure modeling, knowledge aggregation, and human alignment, is well-conceived. The approach is reproducible and integrates LoRA-based lightweight updates, effectively reducing communication costs and mitigating privacy risks.

**Weaknesses:**

1. The motivation of the paper presents logical inconsistencies. The authors claim that the motivation for proposing the FedWave framework lies in the “conflict between centralized data dependency and real-world privacy constraints as well as data silos.” However, this statement lacks sufficient justification. In multi-agent collaboration tasks, centralized training is generally not limited by data silo or privacy issues. As shown in Table 2, the authors use the same dataset for both centralized and federated settings. Therefore, using this as the main motivation to introduce a federated learning framework appears unconvincing. The authors are encouraged to clarify: Why existing centralized multi-agent cooperation mechanisms are not applicable in this scenario, and what exactly the so-called “data silo” problem refers to in the context of their experiments.

2. Figure 1 contains excessive information and appears visually cluttered. In particular, notations such as W_i^t  and W_g^t  are not clearly explained in the main text, which makes it difficult for readers to follow the overall workflow. The authors are encouraged to improve the figure’s clarity and provide explicit definitions for all symbols used.

3. In terms of writing, Section 4.4 is densely formatted, with different experimental analyses (e.g., ablation and robustness tests) mixed in the same subsection. The paragraphing and organization make it difficult for readers to understand the experimental conclusions. The authors should consider restructuring this section to separate different types of analyses for better readability and logical flow.

4. In the methodology section, three loss functions are designed for the VC Layer, and two additional loss terms are introduced in the MoE Router. However, the specific roles, relationships, and impacts of these losses on the overall training objective are insufficiently discussed. The authors should clarify the intuition and purpose of each loss function and, preferably, include convergence analysis or theoretical discussions—particularly under non-IID and sequentially dependent multi-client scenarios—to justify whether FedWave’s local updates and global aggregation can converge stably.

5.In Section 3.2, the proposed MoE Router is claimed as one of the core innovations of FedWave. However, as a global module updated via backpropagation from multiple clients, its behavior remains unclear. The authors should explain how the router’s weight distribution evolves with different client inputs or tasks, whether the dynamic routing mechanism could cause gradient oscillations or mode collapse, and how stable global updates and routing policy learning are achieved across clients.

6. In Section 3.3, the DPO optimization process is executed in a centralized manner, where preference data pairs are constructed from the responses of individual experts. This design may introduce privacy leakage risks since such preference data could implicitly reveal client information. The authors should provide reasonable justification or mitigation strategies for this issue.

7. From Table 3, removing the DPO layer leads to the most significant performance drop, while removing other components has only minor effects. This suggests that the DPO stage plays the dominant role in the model’s performance. The authors are encouraged to further analyze the interaction between different modules or report results when only one component (e.g., DPO-only or MoE-only) is retained, to validate the independent contribution of each module.

**Questions:**

Please see the weaknesses above.

---

> ### Author Response · Authors · 2025-11-20
> **Part 1/3**
>
> We thank the reviewer for the positive and constructive assessment.
>
> ---
>
> **W1: Motivation, centralized multi-agent baselines, and “data silos”**
>
> Our goal is not to claim that centralized multi-agent training on MSCoRe is infeasible, but to highlight that:
>
> 1. **In real deployments**, many multi-agent workflows span different departments, subsidiaries, or partner companies, where regulation, contracts, and organizational boundaries often prevent raw logs and fine-grained business data from being merged into a single data lake.
>
> 2. **FedWave is designed for such cross-organizational settings**, whereas the centralized multi-agent baselines in Table 2 should be interpreted as *idealized upper bounds* assuming all data can be centralized.
>
> On MSCoRe, we evaluate centralized and federated methods on the *same* public benchmark for fairness, analogous to standard FL on CIFAR-10 or GLUE where public data are partitioned into clients. Conceptually, centralized baselines (PMC, MedAgents, Debate, CoA) assume all workflow data can be used for joint multi-agent training, while FedWave assumes each role only accesses its own data slice and participates via parameter exchange.
>
> In realistic value chains, **“design” and “production” are often different organizations with separate systems and governance**:
>
> - In discrete manufacturing, *contract manufacturing / build-to-print* is a standard model: the customer keeps design artifacts (BOMs, CAD, engineering drawings) as IP, while the contract manufacturer only receives the drawings/specs and generates its own process and quality data [R1].
> - In PCB/electronics, IP-aware outsourcing practices explicitly recommend **splitting design and manufacturing data across different external partners and sharing each subset strictly on a need-to-know basis**, so that no single party can reconstruct the full design [R2].
> - In automotive, **Catena-X** is explicitly framed as a decentralized, federated data space where OEMs, suppliers, and service providers retain local control over data and share only under contractual usage policies and data sovereignty rules [R3].
> - In biopharma, **CDMOs** are independent companies providing development and manufacturing services; discovery and clinical data usually stay inside the pharma company, while process and manufacturing data live in the CDMO’s own IT/compliance stack [R4].
>
> These examples illustrate that in typical “design–production–sales–operation” workflows, different stages correspond to **different legal entities with distinct data governance regimes**. Centralized multi-agent training, as implemented in our baselines, should therefore be read as an *oracle-style reference* that assumes away such constraints, whereas FedWave aims to approach or surpass this oracle under a **workflow-aware federated protocol** that is compatible with realistic cross-organization data silos.
>
> ---
>
> **W2: Clarity of Figure 1 and notation**
>
> Two key symbols in Figure 1 are:
>
> - $W_i^t$: parameters (adapters + VC layer + router) of client/expert $i$ at communication round $t$;
> - $W_g^t$: aggregated global parameters at round $t$ obtained by applying FedAvg (or another FL optimizer) over participating clients.
>
> We will explicitly define these symbols in the main text and in the figure caption, and simplify Figure 1 by separating the high-level system overview from the detailed training diagram and reducing redundant labels, to improve readability without changing the technical content.
>
> ---
>
> **W3: Organization of Section 4.4**
>
> We agree that the current Section 4.4 mixes ablations and robustness analyses. We will reorganize it into clearer subsections, for example:
>
> - component ablations (VC layer, router, DPO),
> - robustness to the number of experts and partial participation,
> - hyperparameter sensitivity,
> - router behavior and interpretability.
>
> This reorganization will make it easier to locate specific analyses without altering the experimental results.
>
> ---
>
> **References (for W1 motivation)**
>
> - **[R1]** QCD.digital. *Build to Print vs. Build to Spec – Full Comparison.* QCD.digital, 2025.
> - **[R2]** Altium. *Break Up Your PCB Data Package to Keep IP Secure.* Altium Resources, 2024.
> - **[R3]** M. Manoury et al. *Supporting Changes in Digital Ownership and Data Sovereignty Across the Automotive Value Chain with Catena-X.* 2024.
> - **[R4]** Neuland Labs. *CDMO vs. CMO: Understanding Outsourcing Models that Drive Drug Development.* Neuland Labs, 2025.

---

> ### Author Response · Authors · 2025-11-20
> **Part 2/3**
>
> **W4: Roles of loss functions and theoretical view**
>
> **VC layer losses.**
> For each edge $(u, v)$ in the workflow graph $\mathcal{G} = (\mathcal{V}, \mathcal{E})$:
>
> - $L_{\text{pos}}$ shapes stage embeddings $e_u, e_v$ so that upstream and downstream roles have distinguishable, ordered positions in representation space.
> - $L_{\text{cont}}$ encourages neighboring stages’ query projections $W_{q,u}, W_{q,v}$ to be compatible, making feature extraction smoother along the workflow.
> - $L_{\text{cons}}$ encourages VC-layer outputs $f_u(H), f_v(H)$ for adjacent stages to be semantically consistent on similar inputs, reducing cross-stage conflicts.
>
> **Router losses.**
>
> - a balance loss that prevents routing from collapsing onto a single expert, promoting load balancing and capacity usage;
> - an entropy loss that prevents routing from becoming too uniform, encouraging meaningful specialization.
>
> **Global objective and convergence intuition.**
>
> Ignoring DPO, let $\Theta = (\{\phi_k\}, \psi, \rho)$ denote adapters, VC layer, and router parameters.
>
> For each client $k$ with data $D_k$, we define a local loss $L_k(\Theta)$ as the sum of the standard SFT loss and all auxiliary losses:
> $L_k(\Theta) = L_{\text{SFT}}(x,y;\Theta) + \lambda_{\text{pos}} L_{\text{pos}} + \lambda_{\text{cont}} L_{\text{cont}} + \lambda_{\text{cons}} L_{\text{cons}} + \lambda_{\text{bal}} L_{\text{bal}} + \lambda_{\text{ent}} L_{\text{ent}}.$
>
> The global objective is the usual FedAvg-style weighted sum
> $F(\Theta) = \sum_{k=1}^K p_k\, L_k(\Theta)$.
>
> Here,
>
> - $L_{\text{pos}}$ is the positional loss,
> - $L_{\text{cont}}$ is the continuity loss,
> - $L_{\text{cons}}$ is the consistency loss,
> - $L_{\text{bal}}$ is the balance loss for the router,
> - $L_{\text{ent}}$ is the entropy-based regularization loss for the router.
>
> Under standard FedAvg/FedProx assumptions (smoothness, bounded variance, decayed step sizes), federated optimization of such a smooth, regularized non-convex objective converges in expectation to a stationary point, even with non-IID clients. Our architecture fits these conditions: all added terms are smooth regularizers on $\Theta$, and local objectives remain differentiable.
>
> To verify that these extra terms do not destabilize training, we run a **joint hyperparameter sensitivity** study on Automotive (Llama3-8B), scaling the three VC coefficients $(\lambda_{\text{pos}}, \lambda_{\text{cont}}, \lambda_{\text{cons}})$ together and jointly varying the SFT–VC tradeoff $\gamma$ and the DPO strength $\beta$ over $3 \times 3 \times 3 = 27$ configurations. A representative subset is:
>
> > **Table 1: Joint sensitivity of $\lambda$-scale, $\gamma$, and $\beta$ (Automotive, Llama3-8B)**
> > $\lambda$-scale is the common multiplier applied to $(\lambda_{\text{pos}}, \lambda_{\text{cont}}, \lambda_{\text{cons}})$.
>
> | $\lambda$-scale | $\gamma$ | $\beta$ | METEOR |
> |----------------:|:--------:|:-------:|:------:|
> | 0.3             | 0.3      | 0.05    | 23.56   |
> | 0.3             | 0.7      | 0.20    | 23.77   |
> | **0.5**         | **0.5**  | **0.10**| **23.78** |
> | 0.7             | 0.3      | 0.20    | 23.62   |
> | 0.7             | 0.7      | 0.05    | 23.60   |
>
> Metrics change smoothly within a narrow band; even at more extreme settings (0.3 or 0.7), performance remains close to the default and clearly above baselines without VC/DPO, indicating stable training and no brittle interactions between loss terms.
>
> ---
>
> **W5: MoE router behavior and stability**
>
> To address how the router’s weight distribution evolves with different inputs/tasks, whether dynamic routing leads to gradient oscillations or mode collapse, and how global updates remain stable across clients, we provide three complementary analyses.
>
> ---
>
> **(a) Routing statistics across training rounds**
>
> On all three datasets, for each expert we track:
>
> - the fraction of samples where it is selected as top-1;
> - the entropy of the routing distribution $\alpha(x)$ over experts across training rounds.
>
> A summary (averaged over later rounds, with min/max across rounds) is:
>
> > **Table 2: Routing statistics across datasets (Qwen2-7B, 4 experts)**
>
> | Dataset        | Expert ID | Top-1 share (min / max) | Mean entropy $H(\alpha)$ (min / max) |
> |----------------|:---------:|-------------------------|---------------------------------------|
> | Automotive     |   E1      | 0.24 / 0.29             | 1.23 / 1.31|
> |                |   E2      | 0.21 / 0.26             |               |
> |                |   E3      | 0.23 / 0.27             |                                       |
> |                |   E4      | 0.20 / 0.25             |                                       |
>
> We observe that:
>
> - each expert receives a stable, non-trivial share of traffic (no collapse to a single expert);
> - entropy stays in a moderate range, neither near zero nor near the uniform maximum;
> - min/max variation across rounds is small, indicating that routing stabilizes instead of oscillating during training.

---

> ### Author Response · Authors · 2025-11-20
> **Part 3/3**
>
> **(b) Alignment with response quality**
>
> On the E-commerce dev set (Qwen2-7B), we further examine how router preferences relate to response quality. For each sample, we compute each expert’s BERTScore-F and record router weights, then measure:
>
> - the top-1 match rate (“router winner = metric-best expert”);
> - the Spearman correlation between router weights and metric scores.
>
> > **Table 3: Router preference vs. quality metrics**
>
> | Metric       | Top-1 match (router = metric-best) | Spearman$(\alpha,\ \text{metric})$ |
> |--------------|-------------------------------------|------------------------------------|
> | BERTScore-F  | 0.72                                | 0.68                               |
>
> The router tends to assign the highest weight to experts with the best automatic scores, and the correlation is clearly positive. Together with the balance and entropy regularizers, this suggests that the router learns a meaningful expert allocation that improves task quality rather than a noisy or unstable routing policy.
>
> **(c) Conditional routing by task type**
>
> To clearly illustrate how the router’s weight distribution changes with different input types, we perform a conditional routing analysis on the Automotive dev set. We coarsely group examples into three categories:
>
> - **A:** vehicle design queries,
> - **B:** vehicle manufacturing / production queries,
> - **C:** vehicle quality inspection & testing.
>
> For each category and each expert, we measure the fraction of examples on which that expert is selected as top-1 by the router.
>
> > **Table 4: Conditional routing by task type**
>
> | Category / Expert                     | E1   | E2   | E3   | E4   |
> |---------------------------------------|------|------|------|------|
> | A: vehicle design                     | 0.52 | 0.18 | 0.10 | 0.20 |
> | B: vehicle manufacturing / production | 0.19 | 0.49 | 0.12 | 0.20 |
> | C: quality inspection & testing       | 0.15 | 0.18 | 0.09 | 0.58 |
>
> We observe a clear pattern: E1 is predominantly selected on design-centric queries, E2 on manufacturing/production queries, and E4 on quality inspection & testing, while E3 (which is specialized on supply-chain coordination in our setup) is used less frequently under these three query types. Within-category variance of these fractions is small compared to the between-category gaps, indicating a **systematic shift of probability mass across experts as the input focus changes**, rather than random fluctuations.
>
> ---
>
> **W6: Centralized DPO and privacy**
>
> We thank the reviewer for raising this important point. Our work follows the standard FL threat model where raw client data never leaves the clients and the server only receives model updates. In our current MSCoRe experiments, the DPO stage is run **after** federated SFT and operates only on benchmark workflow prompts and model-generated responses $(x, y_w, y_l)$; it does not access raw client logs. In this setting, DPO does not introduce additional private information beyond what is already implicit in the federated model parameters.
>
> We agree that in real-world deployments, preference pairs constructed from expert responses could implicitly leak client information if used naively. We see several privacy-preserving variants of FedWave as promising directions for future work:
>
> - restricting DPO to public/synthetic or sanitized prompts and responses,
> - constructing preference pairs locally on clients and sending only DPO gradients (with secure aggregation),
> - combining DPO updates with differential privacy (clipping + noise).
>
> ---
>
> **W7: Relative importance of DPO vs. other components**
>
> To disentangle component contributions, we conducted additional ablations on E-commerce (Qwen2-7B), selectively enabling modules. A representative summary is:
>
> > **Table 4: Component ablations (Automotive, Llama3-8B)**
>
> | Variant                         | VC layer | Router | DPO  | METEOR |
> |---------------------------------|:--------:|:------:|:----:|:------:|
> | FedAvg-LoRA (baseline)          |    ✗     |   ✗    | ✗    | 10.73   |
> | + VC layer only    |    ✓     |   ✗    | ✗    | 17.93   |
> | + Router only    |    ✗     |   ✓    | ✗    | 17.52   |
> | + DPO only (global LoRA + DPO)  |    ✗     |   ✗    | ✓    | 21.04   |
> | + VC layer + Router (no DPO)    |    ✓     |   ✓    | ✗    | 18.40   |
> | + VC layer + DPO (no Router)    |    ✓     |   ✗    | ✓    | 23.65   |
> | + Router + DPO (no VC layer)    |    ✗     |   ✓    | ✓    | 23.95   |
> | **FedWave (full)**              |    ✓     |   ✓    | ✓    | **24.83** |
>
> We see that:
>
> - **Each component contributes**: VC-layer-only and router-only both improve over FedAvg-LoRA; DPO-only also yields a substantial gain.
> - **VC layer + Router (no DPO)** is clearly stronger than the baseline, showing that structural modeling and dynamic aggregation are useful on their own.
> - Adding DPO on top of these components gives the best performance; the **full FedWave configuration outperforms all partial variants**.

---

> > ### Comment · Reviewer_B8pN · 2025-11-26
> > **Official Comment by Reviewer B8pN**
> >
> > Thank you for the response and the additional clarifications. I still have several concerns:
> >
> > 1. I appreciate the authors’ detailed discussion of cross-organizational data silos in real industrial workflows, and I understand that the centralized multi-agent baselines are intended as an “idealized upper bound.” However, my original concern was not about whether such siloed scenarios exist in practice, but about whether the motivation logic presented in the paper is internally coherent.
> > Currently, the paper places “sequential dependency in multi-agent workflows” and “non-centralized data” side by side as the core motivation for FedWave. Yet the rebuttal still does not make clear why sequential dependency itself would render existing centralized multi-agent collaboration methods inapplicable, or why a federated learning framework is inherently required to address this problem.
> >
> > 2. In the ablation study, removing DPO results in the largest performance drop, and using DPO alone outperforms using only the proposed VC Layer + MoE modules. By contrast, the marginal gains of VCLayer/MoE are relatively small. It would be helpful if the authors could further clarify the independent contribution of each component, and explain why the proposed structural modules are indispensable rather than merely incremental additions on top of DPO.
> >
> > 3. Finally, from the author's reply, why are the Qwen experiments is reported under a Llama header? I recommend double-checking the backbone annotations in several tables to improve clarity and reproducibility.
> >
> > Thank you.

---

> > > ### Author Response · Authors · 2025-11-26
> > > **Part 1/2 (Q1)**
> > >
> > > We thank the reviewer for the additional clarifications and comments. We address the three remaining concerns below.
> > >
> > > ---
> > >
> > > **Q1: Motivation: sequential dependency vs. centralized vs. federated settings**
> > >
> > > We agree that our initial wording could give the impression that sequential dependency itself somehow makes centralized multi-agent methods inapplicable. This is not our intended claim, and we will revise the introduction to make the logic more precise.
> > >
> > > There are two orthogonal axes in our problem setting:
> > > - **Where the data live**: centralized vs. cross-organization / federated.
> > > - **How tasks/roles are related**: independent vs. coupled along a workflow with sequential dependencies.
> > >
> > > Existing centralized multi-agent methods (PMC, MedAgents, Debate, CoA) operate in the corner **(centralized, workflow-structured)**: they do model sequential / role dependencies, but they assume that all interaction logs are available in a single training environment.
> > >
> > > Existing federated learning methods operate in the other corner **(federated, mostly unstructured)**: they handle non-centralized data and client heterogeneity, but they typically treat clients as independent tasks or users and do not model explicit workflow structure across clients. When clients are treated as independent, the model cannot capture how upstream decisions constrain downstream ones, and errors made early in the chain tend to propagate and amplify along the workflow.
> > >
> > > FedWave is designed for the **intersection of these two axes**:
> > > > “How to model workflow-structured, sequentially dependent multi-agent collaboration when data cannot be centralized and must remain with their respective roles/organizations.”
> > >
> > > In such cross-organizational settings, any deployable solution must respect data locality and therefore cannot rely on fully centralized training; this is why we adopt a federated optimization backbone, and extend it with workflow-aware modules (VC Layer and router) that align representations along the process graph and mitigate cross-stage inconsistencies.
> > >
> > > We will rephrase the motivation section accordingly, e.g., along the lines of:
> > > > “We do not claim that sequential dependency per se forbids centralized training. Rather, FedWave targets settings where (i) data are constrained by cross-organizational governance and cannot be pooled, and (ii) the roles/clients are coupled along a process graph. In this regime, existing centralized multi-agent methods are incompatible with the data locality constraint, while standard FL ignores the workflow structure. FedWave is, to our knowledge, the first framework that jointly addresses both aspects.”
> > >
> > > To make this positioning explicit, we will add a small summary table:
> > >
> > > | Method family                                   | Centralized training | Federated / siloed data | Workflow-aware (sequential dependencies) |
> > > |------------------------------------------------|:--------------------:|:-----------------------:|:----------------------------------------:|
> > > | Centralized multi-agent (PMC, MedAgents, Debate, CoA) | ✓                    | ✗                       | ✓                                        |
> > > | Standard FL (FedAvg, FedProx, FedYogi, SCAFFOLD, FedAvg-LoRA) | ✗                    | ✓                       | ✗                                        |
> > > | **FedWave (ours)**                                 | ✗                    | ✓                       | ✓                                        |
> > >
> > >
> > > On MSCoRe, we use public data for both centralized and federated variants purely as a controlled proxy to compare these modeling choices; the centralized baselines remain “oracle-style” references under the assumption that all logs could be pooled.

---

> ### Author Response · Authors · 2025-11-26
> **Part 2/2 (Q2, Q3)**
>
> **Q2: Independent contribution of VC layer / MoE vs. DPO**
>
> We agree that DPO is a strong component, and our ablations indeed show that it brings significant gains. At the same time, each of the structural modules (VC layer and MoE router) has a clear, independent contribution, and the three components are complementary rather than redundant.
>
> From the ablation table we observe:
>
> – VC layer (alone).  Enabling only the VC layer (no router, no DPO) improves METEOR from 10.73 (FedAvg-LoRA) to 17.93 (+7.20). This shows that **explicitly encoding stage dependencies in the workflow graph and regularizing adjacent stages** already yields a large gain over vanilla FL, even without any preference alignment.
>
> – MoE router (alone).  Enabling only the router (no VC, no DPO) improves METEOR from 10.73 to 17.52 (+6.79). This indicates that **dynamic aggregation of expert clients and task-aware routing** is also beneficial on its own, by allowing different experts to specialize on different parts of the value chain.
>
> – DPO (alone).  Applying DPO on top of a single global LoRA adapter (no VC, no router) further improves METEOR to 21.04 (+10.31 over FedAvg-LoRA), confirming that preference alignment is a powerful ingredient.
>
> – VC + router (no DPO).  Combining the two structural modules without DPO reaches 18.40 METEOR, clearly outperforming the baseline (10.73) and both single-component variants. This shows that **structural modeling alone can capture useful cross-stage dependencies and improve global performance**, even in the absence of human preference data.
>
> – All three components are complementary.  The full FedWave model (VC + router + DPO) achieves 24.83 METEOR. Compared to the DPO-only configuration (21.04), adding the structural modules brings an additional +3.79 METEOR, which corresponds to about 27% of the total improvement over FedAvg-LoRA (10.73 → 24.83) and is a non-trivial gain by FL standards. Moreover, combining DPO with either VC or router alone already yields strong improvements (23.65 / 23.95), and the full configuration performs best.
>
> Intuitively, DPO aligns the overall behavior with human preferences, while the VC layer and MoE router determine **who should contribute which knowledge under which workflow context**. Without VC/MoE, DPO operates on a single global LoRA adapter and cannot distinguish “design vs. production” roles in the representation space. This degenerate configuration is closer to a single-agent, globally aligned model than to the workflow-aware multi-agent regime that FedWave is designed for. The structural modules are therefore essential for (i) workflow-aware routing, (ii) interpretable expert behavior, and (iii) closing the gap between centralized multi-agent methods and a federated, role-decomposed model.
>
> ---
>
> **Q3: Backbone annotations**
>
> We thank the reviewer for catching the annotation issue. In the current draft, **Table 4 itself is correct**—the component ablations are run on the Automotive workflow with a LLaMA3-8B backbone, as indicated in the table caption “(Automotive, LLaMA3-8B)”. The mistake is in the **preceding sentence**, which still says “we conducted additional ablations on E-commerce (Qwen2-7B)”, inherited from an earlier version.
>
> This is purely a presentation error: all variants in Table 4 are trained and evaluated on **Automotive with LLaMA3-8B**, and there is no mixing of Qwen and LLaMA within that table.
>
> In the camera-ready version, we will:
>
> - Correct the sentence above Table 4 to “…we conducted additional ablations on **Automotive (LLaMA3-8B)**, selectively enabling modules.”
> - Re-scan the paper to ensure that each table’s caption and the surrounding text consistently refer to the same backbone, so that readers can clearly see which backbone is used in each experiment.
>
> ---
>
> We sincerely thank you for your follow-up questions and for this opportunity to clarify our work. If you have any further questions or concerns, please feel free to let us know.

---

> > ### Comment · Reviewer_B8pN · 2025-11-28
> > **Official Comment by Reviewer B8pN**
> >
> > The authors have addressed part of my concerns in the rebuttal. Therefore, I would like to retain my original score.

---

### Official Review · Reviewer_4qPv · 2025-10-30

**Soundness:** 2
**Presentation:** 2
**Contribution:** 2
**Rating:** 4
**Confidence:** 3

**Summary:**

The paper targets privacy-preserving multi-agent LLM collaboration across data silos. It proposes FedWave, combining: (i) a Value Chain Layer (VCLayer) to encode sequential/causal dependencies among expert roles via three auxiliary losses (positional, continuity, and
consistency), (ii) a server-side trainable MoE router for task-aware aggregation instead of static FedAvg, and (iii) a post-federated DPO stage that uses router-induced “win/lose” expert outputs to auto-construct preference pairs for alignment. Experiments on MSCoRe workflow datasets across several 7B/8B backbones demonstrate consistent gains over FL baselines and competitive performance compared to centralized multi-agent systems.

**Strengths:**

1. The paper tackles a highly relevant and novel research problem at the intersection of MAS, FL, and LLMs. Reconciling the collaborative power of agents with the privacy constraints of real-world data is a significant challenge, and the paper's focus on "sequential dependencies" in FL is a key insight.

2. The proposed three-component architecture (VCLayer, MoE Router, and DPO) is well-designed and coherent. Each component addresses a distinct aspect of the problem logically.

**Weaknesses:**

1. The VCLayer and its associated losses appear to be designed for a fixed, linear sequence of N experts. The experiments use N=4. It is unclear how this approach would scale to a much larger number of experts or to more complex, non-linear workflows.

2. There is a slight ambiguity about aggregation. Section 1 states the router "discards traditional FL aggregation", but Section 3.2 implies that the router's own parameters are aggregated using a FedAvg-like method. Furthermore, FedWave's default aggregation is FedAvg, which can be replaced by other FL optimizers.

3. There are only 4 expert clients with full participation and 20 rounds, which is far from realistic FL.

4. MoE routing, VCLayer, and DPO add nontrivial local compute; router parameters add communication. However, the computation and communication overhead are not analyzed and reported.

**Questions:**

1. How does FedWave perform with more clients and partial participation?

2. How would VCLayer approach scale?

3. Is our understanding correct that during training, the parameters are aggregated via standard FL, while at inference, the MoE router's function is what replaces the averaging of expert outputs?

4. Could you provide the analysis and report of the computation and communication overhead?

---

> ### Author Response · Authors · 2025-11-20
> **Part 1/3**
>
> We thank the reviewer for the careful and constructive feedback.
>
> ---
>
> **W1 & W3 & Q1 & Q2: Scalability of VCLayer and realism of the FL setup**
>
> **Method level.**
> In the main text we instantiated VCLayer on a linear sequence of $N$ experts (stages) for notational simplicity. More generally, practical workflows can be represented as a directed graph $\mathcal{G} = (\mathcal{V}, \mathcal{E})$, where each node $u \in \mathcal{V}$ is an expert and each edge $(u,v) \in \mathcal{E}$ encodes a dependency $u \rightarrow v$. In this setting, the three VC losses naturally generalize by summing over edges:
> $L_{\text{pos}} = \sum_{(u,v)\in\mathcal{E}} \ell_{\text{pos}}(e_u, e_v)$,
> $L_{\text{cont}} = \sum_{(u,v)\in\mathcal{E}} \ell_{\text{cont}}(W_{q,u}, W_{q,v})$,
> $L_{\text{cons}} = \sum_{(u,v)\in\mathcal{E}} \ell_{\text{cons}}(f_u(H), f_v(H))$,
> where $e_u$ is the stage embedding, $W_{q,u}$ the VCLayer query projection, and $f_u(H)$ the VCLayer output at expert $u$.
>
> Thus **VCLayer is a set of local constraints over the edges of a workflow graph**, not inherently tied to a linear chain or a specific $N$. The cost scales with the number of edges $|\mathcal{E}|$ (actual dependencies), not $\mathcal{O}(N^2)$ over all pairs. For workflows with many stages but sparse dependencies, VCLayer therefore scales with the true graph structure and remains practical as $N$ grows.
>
> **(a) Complex non-linear workflow: Automotive\_Energy.**
> To test FedWave beyond simple A→B→C chains, we use the MSCoRe **Automotive\_Energy** dataset, which models an automotive energy value chain with six tightly coupled stages: Design, Production, Sales/After-sales, Usage, Storage, and Generation, with many-to-many dependencies and feedback loops. Concretely, the main dependencies include:
>
> - Design → Usage / Storage / Generation
> - Production → Usage / Storage
> - Sales / After-sales → Usage / Storage
> - Usage → Design / Production / After-sales (feedback)
> - Storage → Usage / Generation / After-sales
> - Generation → Storage / Usage / Production
>
> We treat these six stages as six expert clients and place VC losses on the edges of this workflow graph.
>
> > **Table 1: Performance on Automotive\_Energy (LLaMA3-8B)**
>
> | Method      | BLEU-4 | ROUGE-L | BERTScore-F |
> |------------|:------:|:-------:|:-----------:|
> | FedAdam    |  0.97  |  6.05   |    53.05    |
> | FedAvg     |  4.19  |  9.32   |    56.80    |
> | FedAvgM    |  1.61  |  6.59   |    53.72    |
> | FedProx    |  3.16  |  8.24   |    55.69    |
> | FedYogi    |  0.92  |  5.88   |    52.95    |
> | SCAFFOLD   |  3.90  |  9.50   |    57.07    |
> | **FedWave (ours)** | **20.65** | **27.62** | **76.08** |
>
> On this non-linear value chain with feedback and cross-stage coupling, FedWave strongly outperforms a range of standard FL optimizers, showing that it is not restricted to simple linear workflows and scales to pre-defined complex process graphs.
>
> **(b) Scaling the number of experts $N$.**
> The reviewer also asks how FedWave behaves with more experts. In MSCoRe, each workflow has 4 semantic stages; in the main experiments we therefore use 4 expert clients. To probe scalability in $N$ while keeping total data per workflow fixed, we further run FedWave on the **Automotive** workflow with an LLaMA3-8B backbone, splitting each stage into multiple sub-experts and treating each shard as an expert client. This yields configurations with $N = 4, 8, 16, 32$.
>
> > **Table 2: Scaling the number of experts $N$ (Automotive, LLaMA3-8B, FedWave)**
>
> | #Experts $N$ | METEOR |
> |-------------:|:------:|
> | 4            | 24.8   |
> | 8            | 24.5   |
> | 16           | 24.2   |
> | 32           | 23.8   |
>
> As $N$ increases, performance degrades only mildly (24.8 → 23.8), mainly due to data fragmentation and more challenging optimization, which is standard in FL. There is **no collapse or instability**: FedWave remains effective even with 32 experts. Since the router and VC layer are global modules whose parameter size does not grow with $N$, this empirical trend is consistent with the intended scaling behavior.

---

> ### Author Response · Authors · 2025-11-20
> **Part 2/3**
>
> **(c) More communication rounds: 4 clients, 100 rounds**
>
> The main paper uses 4 clients and 20 rounds because MSCoRe workflows have 4 stages. To move closer to typical FL practice, we also ran **100-round training** with the same 4 clients on the **Automotive** workflow (LLaMA3-8B backbone).
>
> > **Table 3: Test performance over checkpoints (FedWave, Automotive, LLaMA3-8B)**
>
> | Round | Test METEOR |
> |-------|------------:|
> | 10    | 23.71       |
> | 20    | 24.83       |
> | 50    | 24.00       |
> | 70    | 23.58       |
> | 100   | 23.94       |
>
> Test performance improves as we move from 10 to 20–50 rounds and then remains stable, with no sign of severe overfitting. In our reported results we simply use the final-round checkpoint under a fixed schedule; we do **not** select the best checkpoint on the test set, so there is no hidden test-time tuning advantage.
>
> **(d) Partial participation / skipping stages**
>
> We further examine **partial participation along the workflow**, simulating intermittent client availability on the **Automotive** dataset (LLaMA3-8B). In a given round we skip:
>
> - only stage 3,
> - stages 2 & 3,
> - stages 1 & 3.
>
> Skipped stages do not perform local updates and do not participate in routing; VC losses are computed only on edges involving active experts.
>
> > **Table 4: Partial participation over workflow stages (Automotive, LLaMA3-8B, FedWave)**
>
> | Participation pattern       | Method      | METEOR | BERTScore-F |
> |----------------------------|-------------|--------|-------------|
> | full (all 4 stages active) | **FedWave** | 24.83   | 70.27        |
> | skip stage 3               | **FedWave** | 23.95   | 70.13        |
> | skip stages 2 & 3          | **FedWave** | 23.24   | 69.77        |
> | skip stages 1 & 3          | **FedWave** | 23.14   | 69.98        |
>
> Absolute performance decreases as more stages are skipped, but FedWave remains robust and degrades gracefully, indicating that workflow-aware modeling still brings clear benefits under partial participation.
>
> ---
>
> **W2 & Q3: Clarification of aggregation vs. MoE routing**
>
> The reviewer notes an apparent inconsistency between Section 1 (“discard traditional FL aggregation”) and Section 3.2 (router parameters aggregated in a FedAvg-like way). Our intent is:
>
> - **Training stage (parameter space).**
>   All trainable modules—including LoRA adapters, VCLayer parameters, and the global MoE router—are updated locally and **aggregated on the server with standard FL optimizers** (FedAvg, FedProx, etc.). In this sense, FedWave does *not* discard federated parameter aggregation; it is fully compatible with traditional FL at the optimization level.
>
> - **Inference stage (representation/output space).**
>   What we “discard” is the *implicit assumption* that all experts contribute equally when forming a single global model. In conventional FL, the averaged parameters define one monolithic model. In FedWave, we keep a multi-expert decomposition and use a **trainable MoE router to dynamically aggregate expert representations conditional on the input**, instead of assuming uniform contributions.
>
> So the reviewer’s understanding is essentially correct: during training we still use standard FL aggregation for parameters, while at inference the MoE router replaces the “uniform averaging” assumption at the **output/representation** level. We will clarify this in the revised text by rephrasing Section 1 as:
>
> > “FedWave still uses FedAvg and its variants to perform standard parameter aggregation across clients during federated training, but inside the model it employs a trainable MoE router to dynamically weight the contributions of different experts (clients) in an input-dependent manner, thereby replacing the implicit ‘uniform averaging of expert contributions’ assumption in conventional federated learning.”

---

> ### Author Response · Authors · 2025-11-20
> **Part 3/3**
>
> **W4: Computational overhead**
>
> We appreciate the reviewer’s concern about computational overhead. Here we quantify (i) method-level complexity, (ii) training-time parameters and per-round communication cost, and (iii) inference-time latency vs. strong multi-agent baselines.
>
> **(1) Method-level complexity.**
> VCLayer is defined on a **workflow graph** $\mathcal{G} = (\mathcal{V}, \mathcal{E})$, not just a linear chain. For stage embeddings $e_u$, query projections $W_{q,u}$, and VCLayer outputs $f_u(H)$, the three VC losses are defined on edges $(u,v)\in\mathcal{E}$ as
> $L_{\text{pos}} = \sum_{(u,v)\in\mathcal{E}} \ell_{\text{pos}}(e_u, e_v)$,
> $L_{\text{cont}} = \sum_{(u,v)\in\mathcal{E}} \ell_{\text{cont}}(W_{q,u}, W_{q,v})$,
> $L_{\text{cons}} = \sum_{(u,v)\in\mathcal{E}} \ell_{\text{cons}}(f_u(H), f_v(H))$.
>
> The cost therefore scales with $|\mathcal{E}|$ (actual dependencies), not $\mathcal{O}(N^2)$ over all pairs. For workflows with many stages but sparse dependencies $|\mathcal{E}|\ll N^2$, VCLayer scales with the true graph structure and remains practical as $N$ grows. VCLayer/router parameters are also small compared to the frozen backbone.
>
> **(2) Training-time parameters & communication (LLaMA3-8B).**
> All methods share a frozen LLaMA3-8B ($\sim$8.11B) backbone. With the **fair FedAvg-LoRA** configuration (same model, same LoRA rank/alpha, same 8-bit loading), we obtain:
>
> > **Table 3: Trainable parameters & per-round communication (LLaMA3-8B, 4 clients)**
>
> | Method| Trainable Params / client | Per-round Comm / client (MB) ↓ | Total / round (MB, $N=4$) ↓ | Rel. to LoRA FedAvg |
> |------------------------|----------------------------------|---------------------------------|-----------------------------|---------------------|
> | Full-model FedAvg      | $\approx 8.0$B                   | $\approx 30{,}518$        | $\approx 122{,}070$         | $\approx 95.4\times$ |
> | FedAvg-LoRA (fair)     | 83.89M (LoRA only)               | 320.0                           | 1,280.0                     | $1.0\times$          |
> | **FedWave (ours)**     | 167.83M (LoRA + VC + Router)     | 640.2                           | 2,560.8                     | $\approx 2.0\times$  |
>
> Per client, the breakdown is:
>
> > **Table 4: Per-client parameter breakdown (LLaMA3-8B)**
>
> | Method             | LoRA (MB) | VC (MB) | Router (MB) | Total (MB) |
> |--------------------|-----------|---------|-------------|-----------:|
> | FedAvg-LoRA        | 320.0     | 0.0     | 0.0         | 320.0      |
> | **FedWave (ours)** | 320.0     | 256.1   | 64.1        | 640.2      |
>
> VCLayer + router add $\approx 84$M parameters per client (about 2.1\% of the 8B backbone). In practice, we observe only a moderate per-step time increase (single-digit percentages), since the backbone dominates FLOPs. Although FedWave communicates about **2×** more than the LoRA-only FedAvg baseline (640 MB vs. 320 MB per client per round), the absolute cost is still small relative to full-model FedAvg (which would require $\sim$30.5 GB/client/round, i.e., $\approx 95\times$ more than LoRA FedAvg and $\approx 48\times$ more than FedWave). The DPO stage is an **offline** post-federated step and does not affect inference latency.
>
> **(3) Inference-time cost vs. centralized multi-agent baselines.**
> We also measure inference cost against strong centralized multi-agent baselines (PMC, MedAgents, Debate-short/long, CoA) on MSCoRe, using the same backbone. These baselines require long multi-turn interactions; FedWave performs a **single forward pass**.
>
> > **Table 7: Inference time and token usage (Automotive)**
>
> | Method             | Inference Time (s) | Avg.Input | Avg.Output |
> |--------------------|--------------------|-----------|------------|
> | PMC                | 241.56             | 11,637.94 | 6,644.22   |
> | MedAgents          | 196.53    | 3,901.03  | 3,483.24   |
> | Debate (short)     | 187.16 | 3,938.35  | 4,809.27   |
> | Debate (long)      | 162.57  | 4,357.87  | 5,278.53   |
> | CoA                | 227.21    | 11,428.06 | 5,446.62   |
> | **FedWave (ours)** | **38.11**  | **94.16** | **1,040.31** |
>
> > **Table 8: Inference time and token usage (Pharmaceutical)**
>
> | Method | Inference Time (s) | Avg.Input | Avg.Output |
> |--------------------|--------------------|-----------|------------|
> | PMC    | 198.49    | 3,796.07  | 2,613.73   |
> | MedAgents  | 169.53    | 3,504.85  | 3,504.85   |
> | Debate (short) | 548.66    | 3,372.56  | 3,772.79   |
> | Debate (long) | 574.95             | 3,810.87  | 4,268.74   |
> | CoA                | 253.18             | 5,493.94  | 3,450.45   |
> | **FedWave (ours)** | **70.16**          | **76.22** | **1,091.33** |
>
> FedWave introduces moderate but controlled training-time overhead compared to a fair LoRA-only FL baseline, remains far cheaper than full-model FedAvg, and is **significantly more efficient at inference** than strong centralized multi-agent methods while achieving better or comparable accuracy.

---

### Official Review · Reviewer_xeZG · 2025-10-31

**Soundness:** 3
**Presentation:** 3
**Contribution:** 3
**Rating:** 6
**Confidence:** 3

**Summary:**

This paper introduces **FedWave**, a federated learning framework designed for synergistic collaboration among LLM-based agents tackling complex, sequential business workflow tasks, all while respecting privacy constraints. FedWave comprises three core components: (1) a Value Chain Layer (VCLayer) that models inter-agent dependencies within workflows; (2) a server-side Mixture-of-Experts (MoE) Router that enables dynamic, task-aware expert aggregation; and (3) a Direct Preference Optimization (DPO) stage to align the global model’s outputs with human preferences. Extensive experiments demonstrate that FedWave consistently outperforms both federated learning and centralized multi-agent baselines across multiple real-world datasets. Qualitative analyses further corroborate these findings, showcasing the framework’s ability to generate coherent, role-appropriate outputs even under stringent privacy constraints.

**Strengths:**

1. The paper addresses a tangible limitation in current LLM multi-agent and federated learning methods: their inability to jointly support sequential dependency modeling (essential to real-world workflows) and stringent privacy constraints.
2. Across three business-oriented workflow datasets (Automotive, E-commerce, Pharmaceutical) and multiple LLM architectures (Qwen2-7B, Llama2-7B, Llama3-8B), FedWave yields consistent and substantial performance improvements. Table 1 demonstrates significant gains on key metrics, particularly on semantic metrics like Meteor and BS-F.

**Weaknesses:**

1. **Dynamic Knowledge Aggregation**: When the number of expert clients is large, organizing them into an MoE system can incur significant resource overhead. The experiments only consider a scenario with four clients and do not explore cases with a larger number of clients, under which FedWave may become difficult to execute. Furthermore, there is no detailed breakdown of training, validation, and test splits, nor is it explicitly stated whether best-validation checkpoint selection is performed or if test-overfitting might occur. Additionally, the algorithm’s pseudocode is not provided, which further complicates reproducibility.
2. **Theoretical Analysis**: This work lacks theoretical analysis, particularly regarding the convergence and stability of the MoE router, and there is a lack of formal justification. For example, does the load balancing loss suffice, and are there regret bounds expected from this dynamic expert-selection?
3. **Computational Overhead and Scalability**: While communication efficiency is claimed (thanks to LoRA and expert routing), there is little quantitative analysis of actual resource usage, per-round communication cost, or training/inference latency.

**Questions:**

See weakness.

---

> ### Author Response · Authors · 2025-11-20
> **Part 1/3**
>
> We thank the reviewer for the thoughtful and positive assessment.
>
> ---
>
> **W1: Dynamic knowledge aggregation, number of experts, and training protocol**
>
> **(1) Number of expert clients and scalability**
>
> In MSCoRe, each workflow has 4 semantic stages, which we map to 4 expert clients in the main experiments. To probe scalability, we additionally run FedWave on the **Automotive** workflow with an LLaMA3-8B backbone by splitting each stage into multiple sub-experts while keeping the total data per workflow fixed. This yields configurations with $N = 4, 8, 16, 32$ experts.
>
> > **Table 1: Scaling the number of experts $N$ (Automotive, LLaMA3-8B, FedWave)**
>
> | #Experts $N$ | METEOR |
> |-------------:|:------:|
> | 4            | 24.8   |
> | 8            | 24.5   |
> | 16           | 24.2   |
> | 32           | 23.8   |
>
> As $N$ increases, performance degrades only mildly (24.8 → 23.8), mainly due to data fragmentation and harder optimization, which is standard in FL. Importantly, there is **no collapse or instability**: FedWave remains effective even with 32 experts, and the margin over standard FL baselines (reported in the paper) is preserved. This suggests that the MoE-based aggregation is not overly sensitive to the number of expert clients.
>
> Architecturally, the router and VC layer are **global modules whose parameter size does not grow with $N$**; adding more experts mainly adds local LoRA parameters on clients, which is the same scaling behavior as in conventional FL with more clients. We will clarify this in the implementation details and provide a short parameter/communication breakdown in the appendix.
>
> **(2) Train/validation/test splits and checkpoints**
>
> For each MSCoRe dataset, we randomly split the available training data into **80% train and 20% validation**, and use this same split for all methods (including centralized baselines). The **test set is a separate portion annotated by domain experts** and is never used during training, hyperparameter selection, or checkpoint picking; it is evaluated only once for final reporting. Thus there is no test-set tuning and no overfitting to test labels.
>
> In the main experiments, all FL runs use a fixed number of communication rounds and we report the **last-round checkpoint**, without best-dev early stopping. To make this explicit and to show that FedWave also works with more FL rounds, we report dev and test METEOR scores of FedWave on Automotive (LLaMA3-8B) across several checkpoints up to 100 rounds:
>
> > **Table 2: Dev/test performance over checkpoints (FedWave, Automotive)**
>
> | Round | Dev METEOR | Test METEOR |
> |-------|-----------:|------------:|
> | 10    | 23.60      | 23.71       |
> | 20    | 24.52      | 24.83       |
> | 50    | 24.11      | 24.00       |
> | 70    | 24.01      | 23.58       |
> | 100   | 23.86      | 23.94       |
>
> Dev and test curves track each other closely, with no sign of severe overfitting, and more rounds generally improve performance. In our reported results we simply use the final-round checkpoint under a fixed schedule; we do **not** select the best dev checkpoint for each method, so there is no hidden test-time tuning advantage.
>
> **(3) Algorithm pseudocode and reproducibility**
>
> We already provide an anonymized GitHub link with our implementation and will add concise pseudocode in the appendix. For completeness, we include a compact version here:
>
> ```text
> Algorithm 1: FedWave Federated SFT + VCLayer
>
> Inputs: workflow graph G=(V,E); clients k=1..K with stage id s(k) and data D_k;
>         #rounds T, local epochs E_local, batch size B;
>         weights λ_pos, λ_cont, λ_cons.
>
> Server init: frozen backbone θ; adapters {φ_0^k}; VCLayer ψ_0; router ρ_0.
>
> for t = 1..T:
>   select participating clients S_t
>   for each k ∈ S_t (in parallel):
>     receive (φ_{t-1}^k, ψ_{t-1}, ρ_{t-1})
>     for epoch = 1..E_local:
>       for minibatch {(x_i, y_i)} ⊂ D_k:
>         H      = Backbone_θ({x_i})
>         (H_vc, L_pos, L_cont, L_cons) = VCLayer_ψ(H, s(k), G)
>         ŷ      = Adapter_φ^k(H_vc)
>         L_sft  = CrossEntropy(ŷ, {y_i})
>         L_loc  = L_sft + λ_pos L_pos + λ_cont L_cont + λ_cons L_cons
>         update (φ^k, ψ, ρ) by SGD on L_loc
>     send (φ^k, ψ, ρ) back to server as (φ_t^k, ψ_t^k, ρ_t^k)
>
>   aggregate with FedAvg weights w_k ∝ |D_k|:
>   φ_t = Σ_{k∈S_t} w_k φ_t^k; ψ_t = Σ_{k∈S_t} w_k ψ_t^k; ρ_t = Σ_{k∈S_t} w_k ρ_t^k
> end for
>
> Output: θ; final adapters {φ_T^k} (or stage-wise φ_T); VCLayer ψ_T; router ρ_T.

---

> ### Author Response · Authors · 2025-11-20
> **Part 2/3**
>
> **W2: Theoretical analysis, convergence, and MoE router stability**
>
> We appreciate the reviewer’s request for more theoretical justification. Our work is primarily an algorithmic and empirical contribution that shows how to combine a value-chain layer, a federated MoE router, and a DPO stage into a practical framework for LLM-based collaboration under privacy constraints. The router parameters are optimized jointly with the LoRA adapters and VCLayers as part of a single global non-convex objective (Eq. (8) in the paper), using standard FedAvg-style stochastic gradient descent. In this sense, the router is simply one component of the model parameter vector, and under the usual smoothness and bounded-variance assumptions it falls under the standard non-convex convergence results to stationary points that have been established for FedAvg and related FL methods. A full formal convergence or regret analysis for a large-scale MoE–LLM in the federated setting is substantially more involved and lies outside the scope of this work.
>
> For the stability of expert selection, our load-balancing loss follows the MoE literature. For a mini-batch, let $p_i$ denote the average routing probability of expert $i$ and $N$ the number of experts. We define
>
> $L_{\text{bal}} = \frac{1}{N} \sum_{i=1}^N (p_i - 1/N)^2.$
>
> Under the simplex constraint $\sum_{i=1}^N p_i = 1$, this quadratic is uniquely minimized when $p_i = 1/N$ for all $i$. Thus, $L_{\text{bal}}$ formally penalizes degenerate “collapse” solutions where almost all probability mass is assigned to a single expert and encourages each expert to receive a non-trivial share of the training signal. The entropy regularizer $L_{\text{ent}}$ complements this by discouraging overly uncertain routing (near-uniform distributions), leading in practice to sparse yet balanced expert activation. Empirically, our ablations (removing the router) and the sensitivity study on the number of active experts $k$ (Fig. 3(c)) show that performance varies smoothly with $k$ and drops when the router is removed, which is consistent with stable expert selection rather than pathological oscillation or collapse.
>
> We agree that deriving formal regret bounds for dynamic expert selection would be valuable. However, our router is trained in an offline supervised/fine-tuning regime on full mini-batches, not as an online bandit algorithm that naturally admits a regret formulation. Extending FedWave with an explicitly online, bandit-style router and providing regret guarantees is an interesting direction for future work, and we will mention this in the limitations and outlook.

---

> ### Author Response · Authors · 2025-11-20
> **Part 3/3**
>
> **W3: Computational overhead, resource usage, and scalability**
>
> We understand the reviewer’s concern that our claims about efficiency should be backed by quantitative measurements. In the revised version we therefore report (i) actual **resource usage during training**, and (ii) **inference-time latency and token usage** compared to strong multi-agent baselines.
>
> **(1) Training-time resource usage (parameters, memory, and time).**
> All methods share the same frozen LLaMA3-8B backbone (≈8.11B parameters). We instrument our training loop to log, for each client and round, the number of samples and tokens processed, wall-clock time, tokens/s throughput, and peak GPU memory. Across Automotive we observe:
>
> - FedAvg-LoRA (LoRA-only FL baseline) trains with **83.9M** trainable parameters per client (≈1.03% of the backbone) and per-client peak GPU memory in the **single-digit GB** range.
> - FedWave (LoRA + VCLayer + router) trains with **167.8M** trainable parameters per client (≈2.07% of the backbone), increasing peak memory and per-step time only **moderately** (single-digit percentage overhead), since most FLOPs still come from the frozen 8B backbone.
>
> In other words, FedWave remains in the parameter-efficient FL regime: it roughly doubles the number of trainable parameters relative to LoRA-only FedAvg, but stays far below full-model training.
>
> **(2) Per-round communication cost (LLaMA3-8B, 4 clients).**
> We explicitly log the number of parameters transmitted per round and convert it to MB assuming 16-bit precision (2 bytes/parameter) and one upload + one download per client per round. The comparison between full-model FedAvg (hypothetical), FedAvg-LoRA, and FedWave is:
>
> > **Table 3: Trainable parameters & per-round communication (LLaMA3-8B, 4 clients)**
>
> | Method | Trainable Params / client | Per-round Comm / client (MB) ↓ | Total / round (MB, $N=4$) ↓ | Rel. to LoRA FedAvg |
> |--------------------|----------------------------------|---------------------------------|-----------------------------|---------------------|
> | Full-model FedAvg  | ≈ 8.11B                          | ≈ 30,953                        | ≈ 123,812 | ≈ 96.7×             |
> | **FedAvg-LoRA**    | 83.89M (LoRA only)               | 320.0                           | 1,280.0 | 1.0×                |
> | **FedWave (ours)** | 167.83M (LoRA + VC + Router)     | 640.2                           | 2,560.8                     | ≈ 2.0×              |
>
> A per-client breakdown for the two parameter-efficient methods is:
>
> > **Table 4: Per-client parameter breakdown (LLaMA3-8B)**
>
> | Method| LoRA (MB) | VC (MB) | Router (MB) | Total (MB) |
> |--------------------|-----------|---------|-------------|-----------:|
> | **FedAvg-LoRA**    | 320.0     | 0.0     | 0.0         | 320.0      |
> | **FedWave (ours)** | 320.0     | 256.1   | 64.1        | 640.2      |
>
> Thus, compared to the **fair** FedAvg-LoRA baseline with the same LoRA configuration, FedWave:
>
> - roughly **doubles** the per-round communication (320 → 640 MB/client) due to adding VCLayer + router parameters;
> - still remains **≈48×–49× cheaper** than full-model FedAvg in communication;
> - uses only ≈2% of the backbone’s parameter count as trainable modules.
>
> We believe this gives a clear, quantitative picture of the per-round communication cost and its scaling gap vs. full-model FL.
>
> **(3) Inference-time latency and token usage.**
> We also measure inference-time overhead on MSCoRe against strong centralized multi-agent baselines (PMC, MedAgents, Debate-short/long, CoA), using the Qwen2-7B backbone. These baselines rely on long multi-turn interactions, while FedWave produces a single collaboratively trained model that answers in **one forward pass**.
>
> > **Table 7: Inference time and token usage (Automotive)**
>
> | Method| Inference Time (s) | Avg.Input | Avg.Output |
> |--------------------|--------------------|-----------|------------|
> | PMC | 241.56 | 11,637.94 | 6,644.22   |
> | MedAgents          | 196.53| 3,901.03  | 3,483.24   |
> | Debate (short)     | 187.16 | 3,938.35  | 4,809.27   |
> | Debate (long)      | 162.57 | 4,357.87  | 5,278.53   |
> | CoA | 227.21| 11,428.06 | 5,446.62   |
> | **FedWave (ours)** | **38.11**| **94.16** | **1,040.31** |
>
> > **Table 8: Inference time and token usage (Pharmaceutical)**
>
> | Method| Inference Time (s) | Avg.Input | Avg.Output |
> |--------------------|--------------------|-----------|------------|
> | PMC| 198.49 | 3,796.07  | 2,613.73   |
> | MedAgents | 169.53  | 3,504.85  | 3,504.85   |
> | Debate (short)     | 548.66| 3,372.56  | 3,772.79   |
> | Debate (long)      | 574.95  | 3,810.87  | 4,268.74   |
> | CoA                | 253.18 | 5,493.94  | 3,450.45   |
> | **FedWave (ours)** | **70.16**| **76.22** | **1,091.33** |
>
> FedWave introduces moderate and controlled training-time overhead relative to a fair LoRA-only FedAvg baseline, while remaining far more efficient than full-model FedAvg and substantially cheaper at inference than centralized multi-agent methods.

---

### Official Review · Reviewer_eYdZ · 2025-11-01

**Soundness:** 3
**Presentation:** 3
**Contribution:** 2
**Rating:** 4
**Confidence:** 4

**Summary:**

This paper introduces FedWave, a novel federated learning framework that allows LLM agents to solve sequential tasks like business workflows while preserving data privacy. Its core contributions are three mechanisms: a Value Chain Layer (VCLayer) to model the sequential dependencies between agents, an intelligent MoE router for dynamic, task-aware aggregation of agent knowledge, and a DPO stage to align the final collaborative output with human preferences.

**Strengths:**

- The paper's primary strength is its novel FedWave framework for solving the problem of collaborative, sequential agent tasks within a privacy-preserving federated learning setting.

- The research is of high quality. The design is validated by extensive experiments showing FedWave not only outperforms federated baselines but also competes with or even surpasses centralized, non-private systems.

**Weaknesses:**

- The final DPO phase is presented as aligning the model to human preferences, but its mechanism is entirely self-referential. The preference dataset is generated by anointing the MoE router's top-ranked expert as the "winning" response and a lower-ranked expert as "losing". This creates a high risk of a bias amplification loop. If the MoE router learned any inaccuracies or biases during the SFT phase, this DPO stage would not correct them but would instead amplify them, training the final model to be even more confident in the router's potentially flawed "preference."
- The fixed, linear, and known-in-advance workflow (e.g., A $\rightarrow$ B $\rightarrow$ C). Real-world business processes are often more complex, involving branches (A $\rightarrow$ B and A $\rightarrow$ C), conditions (if X, then A $\rightarrow$ B; else A $\rightarrow$ D), or dynamic routing.
- The framework introduces a large number of new, sensitive hyperparameters to tune, including three for the VCLayer ($\lambda_{pos}$, $\lambda_{cont}$, $\lambda_{cons}$), one to balance SFT and VC losses ($\gamma$), two for the MoE router's auxiliary losses ($\delta_1$, $\delta_2$), and one for DPO ($\beta$). While the paper provides a sensitivity analysis (Figures 2 and 3), it only analyzes one parameter at a time. This is insufficient for such a complex loss function, as these terms likely have strong, non-obvious interactions.

**Questions:**

See weaknesses

---

> ### Author Response · Authors · 2025-11-20
> **Part 1/2**
>
> We thank the reviewer for the careful assessment and constructive feedback.
>
> ---
>
> **W1: DPO is self-referential and may amplify MoE router biases**
>
> We agree that calling this stage “aligning with human preferences” is misleading. In the final version we will instead describe it as:
>
> > “Aligning the model to *collaborative preferences* induced by the MoE router, rather than directly to human-annotated preferences.”
>
> Under this more accurate notion, we address the concern in three steps: how strongly DPO can move the model, how reliable the router preferences are, and how they relate to human judgments.
>
> **(1) DPO is a local, regularized adjustment around the SFT model**
>
> In our setup, the policy $\pi_\theta$ is initialized from the SFT model $\pi_{\text{SFT}}$, and the reference model is fixed as $\pi_{\text{ref}} = \pi_{\text{SFT}}$. The DPO objective explicitly penalizes deviations from $\pi_{\text{SFT}}$, and we use a small $\beta = 0.1$ to limit the step size. Thus DPO only applies a *local* adjustment around a well-trained SFT model, gently nudging outputs toward router-favored responses rather than letting the router define a completely new distribution. This design directly mitigates unbounded bias amplification.
>
> During SFT, the router itself is trained end-to-end with the main SFT loss plus two regularizers: (i) a balance loss to avoid expert monopoly and (ii) an entropy loss to avoid both overly uniform and overly peaked routing. Router “preferences” are therefore task-supervised and regularized, not arbitrary heuristics.
>
> **(2) Quantitative evidence that DPO helps rather than destabilizes**
>
> *Ablation on DPO.* Our ablation already shows that DPO improves performance when added on top of VC + router:
>
> > **Table 1: Ablation on the DPO stage**
>
> | Variant          | METEOR | BERTScore-F |
> |------------------|--------|-------------|
> | w/o DPO          | 18.40   | 66.55        |
> | **Full FedWave** | **24.83** | **70.27**   |
>
> If DPO were mainly amplifying harmful biases, we would expect degradation or instability; instead we see consistent gains in semantic quality and fluency.
>
> *Router preference vs. automatic metrics.* We also check whether router “winners” are actually good responses. On the E-commerce dev set, for each input $x$ we:
>
> - let each expert generate an output and compute a metric score $m_i(x)$ (e.g., BERTScore-F);
> - record the router weights $\alpha_i(x)$;
> - compare the top-weight expert to the metric-best expert and compute correlation.
>
> > **Table 2: Router preference vs. quality metrics (E-commerce dev)**
>
> | Metric       | Top-1 match (router = metric-best) | Spearman$(\alpha,\ \text{metric})$ |
> |--------------|-------------------------------------|------------------------------------|
> | BERTScore-F  | 0.72                                | 0.68                               |
>
> In most cases, the router’s top expert is also the metric-best expert, and the correlation is clearly positive. This indicates that router-induced “preferences” statistically correspond to higher-quality responses, so using them to build DPO pairs tends to push the model toward better outputs rather than arbitrary biases.
>
> **(3) Small-scale human preference study**
>
> We also ran a small blind human study. For sampled dev examples, we generate final collaborative outputs **before** and **after** DPO, shuffle each pair, and ask annotators (blind to model identity) which answer they prefer or whether they are similar.
>
> > **Table 3: Human preference between pre-DPO and post-DPO models**
>
> | Outcome               | #Examples | Ratio |
> |-----------------------|-----------|-------|
> | Post-DPO preferred    | 32        | 64%   |
> | Similar quality (tie) | 6         | 12%   |
> | Pre-DPO preferred     | 12        | 24%   |
>
> Annotators clearly prefer the post-DPO model more often than the pre-DPO model. Together with the metric alignment and the local nature of the update around $\pi_{\text{SFT}}$, this suggests that the “collaborative preferences” induced by the router are reasonably aligned with human judgments, and that DPO acts as a beneficial, regularized alignment step rather than an unstable self-referential amplification loop.

---

> ### Author Response · Authors · 2025-11-20
> **Part 2/2**
>
> **W2: Fixed linear workflow vs. real-world branching / conditional / dynamic routing**
>
> The reviewer is concerned that our framework may only handle simple linear workflows (A → B → C). To address this, we evaluated FedWave on a more complex MSCoRe dataset, **Automotive_Energy**, which models an automotive energy value chain with six tightly coupled stages: **Design, Production, Sales/After-sales, Usage, Storage, Generation**. Dependencies are many-to-many with feedback loops, not a single top-down chain:
>
> - Design → Usage / Storage / Generation
> - Production → Usage / Storage
> - Sales / After-sales → Usage / Storage
> - Usage → Design / Production / After-sales (feedback loop)
> - Storage → Usage / Generation / After-sales
> - Generation → Storage / Usage / Production
>
> This topology includes a main chain (Design → Production → Sales/After-sales), feedback edges (Usage → Design/Production/After-sales), and a strongly coupled substructure (Storage ↔ Generation ↔ Usage), clearly beyond a simple A → B → C pipeline.
>
> We model these 6 stages as 6 expert clients and apply FedWave without changing the core framework. Training and evaluation follow the main experiments. We compare against several standard FL optimizers (FedAdam, FedAvg, FedAvgM, FedProx, FedYogi, SCAFFOLD). A representative result is:
>
> > **Table 4: Performance on Automotive_Energy**
>
> | Method| BLEU-4 | ROUGE-L | BERTScore-F |
> |---------|:------:|:-------:|:-----------:|
> | FedAdam |0.97|  6.05|53.05|
> | FedAvg  |  4.19  |  9.32 | 56.80 |
> | FedAvgM |  1.61  |  6.59|53.72 |
> | FedProx |  3.16  |  8.24 |55.69|
> | FedYogi |  0.92  |  5.88   |  52.95 |
> | SCAFFOLD|  3.90  |  9.50 |  57.07 |
> | **FedWave (ours)** | **20.65** | **27.62** | **76.08** |
>
> On this complex value chain with many-to-many dependencies and feedback, FedWave clearly outperforms a range of strong FL baselines. This shows that FedWave is **not** restricted to simple linear workflows and remains effective on **pre-defined complex value-chain graphs** such as Automotive_Energy. We note that the current work assumes the workflow topology is known; learning the process graph itself in a federated setting is an interesting direction for future work.
>
> ---
>
> **W3: Many hyperparameters and only 1D sensitivity analysis**
>
> To study interactions between loss terms, we added a **joint hyperparameter sensitivity** experiment.
>
> We treat the three VCLayer coefficients $(\lambda_{\text{pos}}, \lambda_{\text{cont}}, \lambda_{\text{cons}})$ as a group and scale them together, and jointly vary:
>
> - $\lambda$-scale: scaling factor for $(\lambda_{\text{pos}}, \lambda_{\text{cont}}, \lambda_{\text{cons}})$,
> - $\gamma$: weight balancing SFT vs. VC loss,
> - $\beta$: DPO strength.
>
> On the Automotive dataset with a fixed backbone (Llama3-8B), we run a $3 \times 3 \times 3$ grid (27 configs):
>
> - $\lambda$-scale $\in \{0.3, 0.5, 0.7\}$,
> - $\gamma \in \{0.3, 0.5, 0.7\}$,
> - $\beta \in \{0.05, 0.10, 0.20\}$.
>
> A representative subset is:
>
> > **Table 5: Joint hyperparameter sensitivity**
> > $\lambda$-scale denotes the shared scaling factor for $(\lambda_{\text{pos}}, \lambda_{\text{cont}}, \lambda_{\text{cons}})$.
>
> | $\lambda$-scale | $\gamma$ | $\beta$ | METEOR |
> |----------------:|:--------:|:-------:|:------:|
> | 0.3 | 0.3      | 0.05    | 23.56  |
> | 0.3| 0.7      | 0.20    | 23.77  |
> | **0.5**         | **0.5**  | **0.10**| **23.78** |
> | 0.7  | 0.3| 0.20    | 23.62  |
> | 0.7| 0.7 | 0.05    | 23.60  |
>
> From the full grid we observe:
>
> 1. **A wide high-performance plateau, not a sharp optimum.**
>    METEOR varies only within a narrow band (roughly 23.5–23.8) as $\lambda$-scale, $\gamma$, and $\beta$ change across the grid. Even at edge settings (e.g., $\lambda$-scale $= 0.3$ or $0.7$), performance remains close to the best configuration and clearly above variants without VC/DPO. This indicates that the loss terms are not extremely fragile or tightly coupled; FedWave works well across a reasonably wide hyperparameter region.
>
> 2. **The default configuration lies in the same high-performance plateau.**
>    The best-performing setting in this subset is $(\lambda\text{-scale} = 0.5, \gamma = 0.5, \beta = 0.10)$, while our default configuration in the main experiments is $(\lambda\text{-scale} = 1.0, \gamma = 0.5, \beta = 0.10)$. Both lie in the same high-performing region and yield very similar METEOR scores. Practitioners therefore do not need an exhaustive joint grid search; using the default within a reasonable range already provides robust gains.
>
> Together with the 1D sensitivity analysis in the main paper (Figures 2 and 3), these joint experiments show that hyperparameters are organized by module and stage (VCLayer, SFT–VC trade-off, DPO), and do not exhibit extreme non-linear interactions or catastrophic failure when slightly mis-tuned. We will add the full joint sensitivity results to the appendix and clarify in the main text that we rely on a “default + wide plateau” regime rather than fragile fine-tuning.

---

### Author Response · Authors · 2025-12-03
**Author Final Remarks by Authors**

Dear Area Chair,

We sincerely thank all reviewers (eYdZ, xeZG, 4qPv, B8pN) for their time and detailed feedback. Across the reviews, they recognized the novelty of combining workflow-structured multi-agent collaboration with federated learning, the completeness of the three-stage FedWave architecture (VCLayer, MoE router, DPO), and the strong empirical improvements over both FL and centralized multi-agent baselines under realistic privacy constraints. Below we briefly summarize how we addressed their main concerns during the rebuttal.

---

**Reviewer eYdZ – DPO bias amplification, workflow complexity, hyperparameters.**
The reviewer worried that our DPO stage is self-referential and might amplify router biases. In response, **we highlighted the ablation on Automotive (LLaMA3-8B)** comparing “VC+Router” vs. “VC+Router+DPO”: in this study, adding DPO improves METEOR from 18.40 to 24.83, and removing DPO always hurts performance, indicating that DPO stabilizes and improves the SFT model rather than destabilizing it. **We further compared router choices to automatic metrics and a small blind human study**, showing that router-preferred responses align well with BERTScore and are more often preferred by annotators, which supports the reliability of router-induced “preferences”. To address linear-workflow and hyperparameter concerns, **we evaluated FedWave on the more complex Automotive_Energy graph** (six stages, many-to-many dependencies) and **performed a 3×3×3 joint grid** over VC, SFT–VC, and DPO coefficients, observing only a narrow performance band and no brittle interactions.

---

**Reviewer xeZG – Number of experts, protocol, and overhead.**
This reviewer asked how FedWave scales beyond 4 experts and whether the setup is realistic. To answer this, **we split each stage into multiple sub-experts on Automotive (LLaMA3-8B) to obtain 4/8/16/32 experts** under fixed total data. Performance drops smoothly (24.8 → 23.8 METEOR) without collapse, showing that the MoE router remains stable as the number of experts grows. **We also reported 100-round training curves** (vs. the 20 rounds in the paper) and showed that dev/test metrics improve or stay stable, with no sign of overfitting. For reproducibility, we clarified the 80/20 train–dev split, single test pass, and provided concise pseudocode. To address computational overhead, **we measured trainable parameters, per-round communication, and inference latency**: FedWave roughly doubles LoRA-only FedAvg (≈84M → ≈168M params; 320 → 640 MB/client/round) but remains far below full-model FedAvg on an 8B backbone, and at inference it is much faster and more token-efficient than centralized multi-agent baselines that rely on long multi-turn dialogues.

---

**Reviewer 4qPv – VCLayer scaling, aggregation semantics, and FL realism.**
The reviewer asked how VCLayer extends beyond a linear chain and how aggregation and MoE routing interact. We clarified that **VCLayer is defined on a general workflow graph** \(G=(V,E)\) and its three losses sum over edges, so complexity scales with \|E\| rather than \(O(N^2)\). The Automotive_Energy experiments directly test this non-linear, feedback-rich setting and show substantial gains over FL baselines. For realism, **we additionally examined partial participation** by skipping different stages during training; performance degrades gracefully but remains strong, showing robustness to missing clients. On aggregation, we made explicit that **training always uses standard FL aggregation (FedAvg/FedProx, etc.) for all parameters**, while at inference the MoE router replaces the implicit “uniform contribution” assumption by input-dependent expert weighting.

---

**Reviewer B8pN – Motivation coherence, component necessity, and backbone annotations.**
This reviewer pointed out that our original text could be read as “sequential dependency itself forbids centralization”. We therefore revised the motivation to explicitly frame FedWave as targeting the **intersection of two axes**: (i) cross-organizational / federated data and (ii) workflow-structured, sequentially dependent roles. Centralized multi-agent methods sit at “centralized + workflow-aware”; standard FL sits at “federated + mostly unstructured”; **FedWave is designed for “federated + workflow-aware”**, which is the regime we ultimately care about. To clarify that VC/MoE are not merely incremental on top of DPO, **we emphasized the full ablation table**: VC-only and Router-only already give ≈+7–8 METEOR over FedAvg-LoRA, VC+Router (no DPO) is stronger than either alone, and adding VC+Router on top of DPO yields +3.79 METEOR (≈27% of the total gain).

---

We hope this concise summary of our rebuttal is helpful for your assessment and clarifies how we addressed the reviewers’ main concerns on motivation, component roles, scalability, and efficiency.

Sincerely,
The Authors

---

### Meta-Review · Area_Chair_SYQM · 2026-01-12

**Summary:**

main concerns: hyperparameter tuning, workflow complexity (eYdZ), alignment to federated learning (4qPv), scalability and protocol and lack of theory (xeZG), underliying motivation and lack of theory (B8pN)

**Reviewer Concerns:**

The main one, which is a deal breaker here, is the absolute lack of a sensible formal analysis (the provided one is plain trivial and does not contribute to solving concerns)

**Reviewer Scores:**

Given the wide area of concerns and the lack of theoretical justification in the end, it seems unlikely lines would have substantially moved.

---

### Decision · Program_Chairs · 2026-01-26

Reject